# High-Grade Serous Ovarian Cancer—A Risk Factor Puzzle and Screening Fugitive

**DOI:** 10.3390/biomedicines12010229

**Published:** 2024-01-19

**Authors:** Jacek Wilczyński, Edyta Paradowska, Miłosz Wilczyński

**Affiliations:** 1Department of Gynecological Surgery and Gynecological Oncology, Medical University of Lodz, 4 Kosciuszki Str., 90-419 Lodz, Poland; 2Laboratory of Virology, Institute of Medical Biology of the Polish Academy of Sciences, 106 Lodowa Str., 93-232 Lodz, Poland; eparadow@cbm.pan.pl; 3Department of Surgical, Endoscopic and Gynecological Oncology, Polish Mother’s Health Center—Research Institute, 281/289 Rzgowska Str., 93-338 Lodz, Poland; jrwil@wp.pl; 4Department of Surgical and Endoscopic Gynecology, Medical University of Lodz, 4 Kosciuszki Str., 90-419 Lodz, Poland

**Keywords:** ovarian cancer, risk factors, diagnosis, prediction, biomarkers, liquid biopsy, circulating tumor cells, circulating tumor DNA

## Abstract

High-grade serous ovarian cancer (HGSOC) is the most lethal tumor of the female genital tract. Despite extensive studies and the identification of some precursor lesions like serous tubal intraepithelial cancer (STIC) or the deviated mutational status of the patients (*BRCA* germinal mutation), the pathophysiology of HGSOC and the existence of particular risk factors is still a puzzle. Moreover, a lack of screening programs results in delayed diagnosis, which is accompanied by a secondary chemo-resistance of the tumor and usually results in a high recurrence rate after the primary therapy. Therefore, there is an urgent need to identify the substantial risk factors for both predisposed and low-risk populations of women, as well as to create an economically and clinically justified screening program. This paper reviews the classic and novel risk factors for HGSOC and methods of diagnosis and prediction, including serum biomarkers, the liquid biopsy of circulating tumor cells or circulating tumor DNA, epigenetic markers, exosomes, and genomic and proteomic biomarkers. The novel future complex approach to ovarian cancer diagnosis should be devised based on these findings, and the general outcome of such an approach is proposed and discussed in the paper.

## 1. Introduction

High-grade serous ovarian cancer (HGSOC) is the most lethal tumor of the female genital tract due to several reasons, mainly a lack of screening programs and delayed diagnosis, high proliferative potential, and chemo-resistance followed by a high recurrence rate. Therefore, despite therapeutic progress, 5-year survival, especially in advanced patient populations (clinical stages III-IV), is invariably low and unsatisfactory (data from American Cancer Society 2020. https://www.cancer.org/cancer/ovarian-cancer/detection-diagnosis-staging/survival-rates.html, accessed on 8 September 2022). Identifying potential etiological risk factors for HGSOC would be a big step in managing this cancer. Although serous tubal intraepithelial cancer (STIC) and “p53 signature” have been recognized as canonical precursor lesions for HGSOC [1], the alternative pathways of cancer development are probable but still obscure. It seems that the alternative path could be the transformation to HGSOC from the ectopic representation of fallopian tube-related lesions on the surface of the ovary (like low-grade serous carcinoma, serous borderline tumor, endosalpingiosis, and the tubal-type ovarian cortical inclusion cyst) [2]. The issue needs further investigation, as different precursor lesions could influence the natural evolution of the tumor and its particular biology. Similarly, the germinal *BRCA* mutation is an exemplary model for HGSOC carcinogenesis; however, we have not been able to answer if it is enough to initiate cancer or if some other trigger factors are needed to push the *BRCA* mutation toward ovarian cancer. The direct contact between the vaginal/external environment and peritoneal cavity in women also creates a unique possibility to modify the abdominal cavity by environmental factors, including bacterial and viral infections. When recognized early, cancer gives a better chance for successful treatment and improved survival. Therefore, attempts to create a useful diagnostic tool for ovarian cancer are ongoing. However, despite several propositions, there is still a considerable gap between diagnostic biomarker discoveries and the practical utilization of some in screening programs. The main two directions in the construction of screening programs should be the identification of a high-risk population of patients followed by an appropriate approach to maintaining fertility and later the use of prophylactic procedures (like risk-reduction bilateral salpingo-oophorectomy—RRSO). Secondly, the screening of the low-risk population is now utterly unavailable for both economic and logistical reasons, but due to the progress in the decline of the costs of laboratory and molecular testing, it could be a more realistic option shortly. It is even more desirable, taking into consideration that the treatment of advanced ovarian cancer consumes the lion’s share of health care resources for gynecological oncology, as well as being associated with highly negative emotional burdens for the patients, their families, and medical professionals. Another problem is the usage of biomarkers for disease monitoring to detect cancer recurrence as early as possible. This review examines the achievements in recognizing the risk factors and identifying both classical and novel biomarkers potentially useful for the construction of screening programs and disease monitoring. In the end, the general outcome of the novel future complex approach to an ovarian cancer diagnosis is proposed and discussed in this paper.

## 2. Risk Factors for HGSOC

### 2.1. Epidemiological Risk Factors

The risk factors that have been commonly accepted and cited in many publications include age (usually less than 45 years), more ovulatory cycles during a lifetime [3], early menarche and late menopause, nulliparity, a lack of breastfeeding [4,5], the non-usage of oral hormonal contraceptives [6], the use of estrogen menopausal replacement therapy [7], diabetes, obesity, and cigarette smoking [3]. The observation that tubal ligation, salpingo-oophorectomy, and hysterectomy could protect against HGSOC suggests the ascending infection and the use of talc on examination gloves or for perineal hygiene as a risk factor [5,8]. A diet rich in animal proteins contains xeno-estrogens, which have carcinogenic potential [9]. High levels of serum lipids resulting from increased fat consumption stimulate leptin secretion by fatty tissue, stimulating the release of gonadotropins. This mechanism could result in disturbed re-epithelialization of ovaries and predisposes one to abnormal epithelium proliferation [10]. A lower intake of polyunsaturated omega-3 fatty acids and a high consumption of trans-fats were also associated with an increased risk of ovarian cancer [11]. However, epidemiological analysis revealed that only about 10% of ovarian cancer cases could be attributed to the presence of the above-mentioned risk factors [12]. 

### 2.2. Gene Mutations

Genetic studies have brought knowledge about other risk factors. The most discussed of them is the presence of the familiar germline *BRCA1/2* mutation. It is estimated that about 3.5% of ovarian cancer patients have germline *BRCA* mutations, and 10–20% of ovarian cancers could be associated with these mutations [13,14]. The risk of ovarian cancer in this group of patients varies from 27% to 44% in the *BRCA2*-mutated and *BRCA1*-mutated patients, respectively [15]. For comparison, the lifetime risk of ovarian cancer in non-mutated populations does not exceed 1.5% [3]. Except for high-penetrance (high-risk) *BRCA* alleles, several low-penetrance (low-risk) alleles could account for the increased risk of ovarian cancer, including mutations in BRCA1-interacting helicase 1 (*BRIP1*), RAD51 Paralog C (*RAD51C*), *RAD51D*, and partner and localizer of BRCA2 (*PALB2*) genes, with the odds ratio of ovarian cancer being 14.1 for *BRIP1*, 5.2 for *RAD1C*, and 12.0 for *RAD1D* mutation and a relative risk of 2.9 for *PALB2*, respectively [16,17]. This risk ratio translates to a lifetime risk of 5–12% [18]. Both high- and low-risk mutations account for an ordinary risk of 40% for ovarian cancer [19]. The association of other low-penetrance allele mutations was also considered for ATM serine/threonine kinase (*ATM*), checkpoint kinase 2 (*CHEK2*), and BRCA1-associated RING domain 1 (*BARD1*) genes. According to the National Comprehensive Cancer Network (NCCN v.2.2022.), only the mutation of the *ATM* gene could influence the increase in the risk of ovarian cancer. It should be considered for familiar screening [20]. All these genes are connected with the homologous recombination (HR) pathway of DNA repair, which is seriously disturbed in HGSOC. Therefore, patients with the presence of one high-risk mutation or at least two low-risk mutations may benefit from therapies targeting dysfunctional genes, like poly (ADP-ribose) polymerase (PARP) inhibitors and drugs of similar mechanisms of action [21]. Lynch syndrome, also called hereditary nonpolyposis colorectal cancer syndrome (HNPCC), is associated with a disturbed DNA mismatch repair system and microsatellite instability due to the MutL protein homolog 1 (*MLH1*) and MutS homolog 2 (*MSH2*) mutations [22]. A clinical sign of the syndrome is colorectal cancer associated with endometrial and ovarian cancers. Ovarian cancer risk in women with Lynch syndrome is 6–8% and is higher in the presence of *MSH2* and *MSH6* mutations [23]. Women with Lynch syndrome-associated ovarian cancer are diagnosed earlier, usually have non-serous histology, and, in 22% of cases, ovarian tumors exist with synchronous endometrial cancer [24,25]. Li–Fraumeni syndrome is an autosomal dominant syndrome determined by heterozygous germline mutations in the tumor suppressor gene *TP53.* Less frequent tumors (15%) associated with Li–Fraumeni syndrome include ovarian cancer and occur at an earlier age compared to the onset age of sporadic ovarian cancer [26,27]. Tumors are typically characterized by worse survival, increased chemo-resistance, and a high relapse rate [28].

### 2.3. Gene Polymorphisms

The analysis of single nucleotide polymorphisms (SNPs) in many genes regulating ovarian cancer biology has been performed. Although the *CHEK2* mutations were not confirmed as significant regarding the risk of ovarian cancer, other studies indicated that *CHEK2* SNP rs17507066 had a significant association with the risk of HGSOC [20]. In a group of 587 studied patients with epithelial ovarian cancer, SNPs at *PIK3CA* rs9838117 and *ERBB2* rs1058808 loci associated with cancer cell signaling were found to be correlated with the risk of cancer [29]. In the panel of genes related to ovarian cancer relapse, SNP analysis indicated that the male-specific lethal-1 homolog (*MSL1*) SNP rs7211770 was associated with a decreased risk of HGSOC (OR 0.81). At the same time, tumor suppressor HEXIM P-TEFb Complex Subunit 1 (*HEXIM1*) SNP rs1053578 was correlated with an increased risk of ovarian cancer (OR 1.4) [30]. The transcription elongation factor A (SII)-like 7 (*TCEAL7*) gene is epigenetically down-regulated in ovarian cancers, and SNPs rs5987515, rs5987724, and rs5945971 were significantly associated with a reduced risk of HGSOC [31]. Micro RNA (miR) miR-196a-2 locus polymorphism is observed in several malignancies, and studies have indicated that the miR-196a-2 rs11614913 CC genotype may increase the risk of ovarian cancer and enhance cell invasion [32]. The hephaestin gene (*HEPH*) encoding the protein responsible for iron transport regulates the proliferation and apoptosis of cancer cells and facilitates reactive oxygen species (ROS) production [33]. The *HEPH* SNP rs17216603 was associated with a reduced risk of HGSOC (OR 0.81) [34]. Somatic mutations in the retinoblastoma-1 (*RB1*) gene have been observed in several malignancies. The *RB1* rs2854344 and rs4151620 polymorphism have been associated with a reduced risk of ovarian cancer [32]. The insulin-like growth factor binding protein 3 (*IGFBP3*) gene functions as a low-penetrance onco-suppressor gene and regulates the interaction between insulin growth factor-1 (IGF-1) and its receptor [35]. IGF-1 enhances tumor growth and inhibits cell apoptosis. The *IGFBP3* SNP rs2270628 was associated with increased IGF1 plasma levels and higher ovarian cancer risk [36]. The epigenetic regulation of cancer growth is, among other things, regulated by DNA methylation provided by DNA methyltransferases encoded by *DNMT* genes. The *DNMT1* rs2228611 and rs759920 SNPs were associated with an increased risk of ovarian cancer development (OR 1.8 and OR 1.9, respectively) [37]. The inflammatory pathways are activated in many cancers and are important in ovarian cancer. The interleukin-1α (*IL1A*) gene SNP rs17561 was associated with a reduced risk of endometroid, mucinous, and clear-cell ovarian cancers but not HGSOC, indicating the genetic diversity of ovarian epithelial tumors [38]. Toll-like receptors (TLRs) are engaged in the inflammatory pathways, and their activation in ovarian cancer is closely related to cancer aggressiveness, chemo-resistance, and adverse clinical outcomes [39,40]. The interaction between TLR4 and MyD88, the molecule considered to be the marker of stem cells in ovarian cancer, contributes to inflammation in the tumor environment and enhances its aggressive phenotype [41]. Recent findings suggest that the *TLR4* Asp299Gly polymorphism could be a genetic risk factor for the development of ovarian cancer [42]. Class II human leukocyte antigens (HLA) regulate interactions between cancer cells and host immune responses represented by tumor-infiltrating lymphocytes (TILs). The intercellular adhesion molecule-1 (ICAM-1) also plays an essential function in immune reactions and cell–cell contact in the tumor environment. The significant association of *HLA-DP* rs3077 AA, *HLA-DQ* rs3920 GG, *ICAM-1* rs1437 CC, and CT genotypes with increased risk of ovarian cancer (OR 43, 6, 25, and 2.6, respectively) was confirmed. Moreover, *HLA-DQ* rs3920 and *ICAM-1* rs1437 alleles varied significantly among borderline and malignant types of ovarian cancer, highlighting once more the need for an individual approach to different ovarian cancer subtypes [43]. Recently, calculating polygenic risk scores for ovarian cancer has improved risk stratification and may be used in clinical prevention programs [44]. 

### 2.4. Microbiome

#### 2.4.1. Viral Infections

It is estimated that microbial infections account for even 20% of cancer cases, and several viruses have been involved in carcinogenesis directly and indirectly through the stimulation of chronic inflammation and immunosuppression [45,46]. Studies confirming the presence of viruses in ovarian tumors have been noticed since 1992 when the presence of human papillomavirus (HPV) 16 and 18 DNA was confirmed in ovarian cancer tissue [47]. Investigations of HGSOC confirmed the presence of HPV 16 and 33 DNA in 10.5% of samples [48], HPV 6 DNA in 25% [49], HPV 16 DNA in 5.7% [50], and cytomegalovirus (CMV) DNA in 40–50% of samples [49,51]. HPV and CMV co-infection was observed in almost two-thirds of malignant ovarian cancer tissues containing mostly HGSOC, and the presence of viruses was also noticed in fallopian tubes [52]. Generally, in advanced ovarian cancer, the predominant genotypes are high-risk HPV 16 and 18, similar to cervical cancer [53]. The integration of HPV oncogenes *E6* and *E7* followed by the degradation of p53 and Rb proteins triggers carcinogenesis [54] and could hypothetically account for the origin of superficial tubal intraepithelial cancer (STIC), which precedes HGSOC. Of the genes that could be affected by HPV 16 insertions, more than half were found to be associated with malignant solid tumors, and in the case of HPV 18, two genes associated with malignancy in solid tumors were affected by genomic integration [55]. Similarly, *Herpes* virus insertions were also noticed at the sites of genes related to epithelial cancer or different solid cancers [55]. The CMV immediate-early (IE) and late tegument (pp65) proteins were abundant in ovarian cancer tissues and correlated with worse prognosis [51]. CMV proteins can inhibit the cycle arrest functions of p53 and modulate the function of many key signaling pathways, including Wnt/β-catenin, PI3K/AKT, NF-κB, and STAT3, thus stimulating proliferation, angiogenesis, inflammation, and immune escape [56,57,58,59]. Epstein–Barr virus (EBV) is a herpes virus associated with several human malignancies [60]. The presence of EBV DNA was confirmed in 7.8% of ovarian cancer samples, but no EBV DNA presence was noticed specifically in HGSOC samples [61,62]. Compared to controls, ovarian cancer is also characterized by a different composition of viral signatures, with *Anelloviridae*, *Astroviridae*, *Birnaviridae*, *Bornaviridae*, *Hepadnaviridae*, *Iridoviridae*, *Paramyxoviridae*, and *Rhabdoviridae* found in significant levels exclusively in ovarian cancer samples [55]. Similarly, the *Poxviridae* family and Merkel cell *Polyomavirus* were also over-represented in ovarian cancer tissues [55]. 

#### 2.4.2. Bacterial Microbiome

In addition to viruses, a bacterial microbiome could account for ovarian cancer risk. Changes to microbiome compartments (such as vaginal, peritoneal, or tumor tissue compartments) may play an essential role in the pathogenesis of ovarian malignancy. Such a shift in the microbiome in the case of neoplastic diseases is termed oncobiosis. Oncobiosis-related interactions may influence cancer hallmarks, i.e., cellular metabolism, or induce gene expression changes, eventually leading to larger-scale events, such as epithelial-to-mesenchymal transition or angiogenesis [63]. Both bacteria and bacterial metabolites modulate the actions of the immune system. The oncobiotic transformation may cause a tolerogenic state that reduces the elimination of early cancer cells. However, immunogenic actions related to the dysbiotic microbiome may also lead to tumorigenic inflammation, the induction of oxidative stress, and a higher risk of mutations [63]. The oncobiotic microbiome in the peritoneum or tumor tissues differs from the healthy state. Several authors have suggested that the immunogenic character of the microbiome is associated with an increase in the presence of Gram-negative bacteria [64,65,66]. 

The observation that a history of pelvic inflammatory disease (PID) increased the risk of ovarian borderline tumors and HGSOC suggested a link between genital tract infection and ovarian cancer [67,68]. Another study showed that a history of more than five episodes of PID was correlated with a 2.5-times more significant risk of ovarian cancer in women aged over 35 years [69]. One of the considerable causes of PID is *Chlamydia trachomatis* infection [70]. The DNA of *Chlamydia* was found in 17% of ovarian cancer samples, and chlamydial heat shock protein-60 (HSP-60) was detected more frequently in HGSOC samples than in benign ovarian tumors [71]. The presence of chlamydial plasmid-encoded Pgp3 (virulence factor in genital tract) antibody was associated with a doubled risk of ovarian cancer [72]. The mechanistic explanation for the relation between *Chlamydia* infection and ovarian cancer is based on the observations that infection provokes oxidative DNA damage, the down-regulation of p53 function, and induces DNA double-strand breaks [73,74]. Moreover, chlamydial infection causes an epithelial-to-mesenchymal transition (EMT) in host cells and increases the stemness and CpG methylation of genes. All of these actions may contribute to promoting HGSOC cancer [75,76]. Menopausal status is one of the factors influencing the composition of cervicovaginal microbiota. In healthy premenopausal women, a *Lactobacillus*-dominated microbiota was observed. At the same time, after menopause, it shifted to a highly diverse bacterial species dominated by *Propionibacterium*, *Corynebacterium*, and *Staphylococcus*, similar to skin bacterial composition [77,78]. In ovarian cancer, a *Lactobacillus*-poor and highly diversified microbiota mimicking the post-menopausal environment was described, regardless of the menopausal status [77]. Surprisingly, it was discovered that women with the *BRCA1* mutation had *Lactobacillus*-depleted cervicovaginal microbiota even before the onset of ovarian cancer [79]. In the study performed by Banerjee et al. [55], two predominant bacterial phyla were detected, mainly *Proteobacteria* and *Firmicutes*, present in 52% and 22% of ovarian cancer samples, respectively. Similarly, the *Proteobacteria*/*Firmicutes* ratio was significantly increased in ovarian cancer samples [80]. This study confirmed that ovarian cancer samples had much more diversified microbiota than controls. 

The upper genital tract was in the past thought to be a sterile environment; however, recent studies contradict these notions. The ovarian follicular fluid of patients subjected to in vitro fertilization contained bacteria such as *Actinomyces*, *Staphylococcus*, *and Bifidobacterium* [81]. The endometrial cavity also indicates the presence of bacteria, with *Bacteroides* being a dominant resident of the non-pregnant endometrium [82]. A microbiome was also confirmed in the fallopian tubes, fimbriae, and ovarian surface epithelium. *Firmicutes*, *Proteobacteria*, *Actinobacteria*, and *Bacteroidetes* phyla were identified in the upper genital tract, with *Proteobacteria* being the most abundant while *Firmicutes* were the least common microbiota, respectively. There were also differences in microbiota distribution between tubes, fimbriae, and ovaries [83]. In ovarian cancer, the most abundant microbiota were *Ralstonia*, *Mycobacterium*, and *Variovorax*, and the abundances of several taxa were reduced in cancer compared to control samples, indicating the occurrence of bacterial dysbiosis in ovarian cancer [83]. 

The probable influence of the gut microbiome on polycystic ovary syndrome and endometriosis or chronic pelvic pain has been postulated [84,85]. The impact of the gut microbiome on the formation and development of ovarian cancer is much more hypothetical; however, the connections between gut microbiome and intestinal estrobolome could explain this dependency. Firstly, the gut microbiome disturbs the entero-hepatic circulation of estrogen [86]. Secondly, the microbiome interferes with the secretion of β-glucuronidase, an enzyme responsible for the degradation of the active forms of estrogen [87]. Through both mechanisms, the gut microbiome up-regulates the estrogen level and induces carcinogenesis with β-glucuronidase [88]. A recent study on a mouse model of HGSOC could confirm the role of the gut microbiome in ovarian cancer. Control and antibiotic-treated mice were stimulated by tamoxifen to build up malignant ovarian tumors. After one year of observation, it was found that antibiotic-treated mice had significantly fewer and less advanced tumors. The changes in gut microbiota following antibiotic therapy included the increase in *Bacteroidales* species, which are negatively correlated with cancer, accompanied by the reduction or elimination of cancer-associated *Prevotella*. It seems that *Bacteroidales* in the fecal microbiota might prevent the recolonization of cancer-associated *Prevotella* and further inhibit HGSOC development [29]. 

#### 2.4.3. Fungal and Parasitic Infections

The fungal and parasitic signatures have also been found to differentiate between ovarian cancer patients and control groups. The signatures of *Cladosporium*, *Pneumocystis*, *Acremonium*, *Malassezia*, and *Pleistophora* were detected in all ovarian cancer samples, whereas signatures of *Rhizomucor*, *Rhodotorula*, *Alternaria*, and *Geotrichum* were detected in 95% of cancer samples [55]. Parasitic signatures in ovarian cancer samples were significantly more diversified compared to controls. They revealed the presence of genetic material from *Dipylidium*, *Trichuris*, *Leishmania*, and *Babesia* in all samples, as well as signatures of *Trichinella*, *Ascaris*, and *Trichomonas* in more than 95% of cancer samples [55]. The risk factors for HGSOC are summarized in Table 1.

## 3. Prediction of HGSOC

### 3.1. Tumor Markers

#### 3.1.1. CA125

The classic and most widely used tumor marker for HGSOC is CA125, a glycoprotein encoded by the *MUC16* gene [89]. CA125 is detected in serum samples of more than 80% of ovarian cancer patients; however, its accuracy for detecting early-stage ovarian cancer is limited [90,91]. The low sensitivity of CA125 (50–60%) for detecting early-stage ovarian cancer arises because about 50% of such patients do not have elevated CA125 levels [92]. The specificity of CA125 is also relatively low (70–80%), and about 6/10 patients with increased CA125 levels do not have ovarian cancer but have different causes, like other malignancies and benign conditions (menstruation, pregnancy, benign ovarian tumors, uterine fibroids, endometriosis, adenomyosis, pelvic inflammatory disease, and liver diseases) [93]. To improve CA125 accuracy, different approaches have been developed. The detection of CA125-containing exosomes was shown to improve specificity, sensitivity, and AUC [94]. Detecting the glycoforms of CA125 is another strategy to enhance the efficacy of CA125 testing. CA125 is a highly glycosylated protein containing many Thomsen-nouveau (Tn) antigens, which are significantly up-regulated in ovarian cancer cells but have low expression in normal tissue. The combined use of CA125-Tn improved the diagnosis specificity in patients over 45 years [95]. Another form of Tn antigen, the sialyl-Tn antigen (STn), is exclusively abundant in cancer cells but has limited concentration in normal cells. The STn serum levels were increased by around 50%, 10%, and 4% in ovarian cancer patients, endometriosis, and healthy controls, respectively [96]. 

#### 3.1.2. HE4

Human Epididymis Secretory Protein 4 (HE4) has been introduced to ovarian cancer diagnosis. Compared with CA125, HE4 is less frequently affected by benign gynecological conditions, especially endometriosis [97]. HE4 provides a specificity of 96% and a sensitivity of 67% for the detection of ovarian cancer [98]. HE4 enabled the diagnosis of more than half of ovarian tumors not expressing CA125. In early-stage ovarian cancer, the sensitivity and specificity of HE4 were 0.64 and 0.87, respectively, and HE4 performed better than CA125 concerning the specificity (97% vs. 67%) [99,100]. However, similarly to CA125, HE4 is not specific to ovarian cancer and may be increased in other malignancies (endometrial, lung, and breast cancers) and adenomyosis [97,99]. The diagnostic accuracy of HE4 is modified by both the age and menopausal status of the patients. Therefore, HE4 performed better in postmenopausal women than in premenopausal women but had decreased specificity and sensitivity in older women [101,102]. As with CA125, impaired renal function could be a source of significant mistakes in HE4 interpretation [102]. 

#### 3.1.3. CA-72-4

Glycoprotein CA72-4 is elevated in ovarian cancer, but contrary to CA125, it is not affected by the menstrual cycle, pregnancy, or endometriosis [103,104]. Its increased levels were detected in mucinous and clear-cell cancers, where HE4 usually was not changed. Therefore, CA72-4 is a valuable additional marker in the cases missed by standard diagnostic markers [105]. 

#### 3.1.4. FOLR1

Folate receptor-α (FOLR1) is over-expressed in several malignant cancers, including 76% of patients with HGSOC [106,107]. Its serum concentration significantly increases in malignant compared to benign ovarian tumors and healthy controls [108]. The concentration of FOLR1 depends on the clinical stage of the cancer and its histology, grade, and size; therefore, FOLR1 serum levels are much higher in the case of aggressive and advanced tumors of the HGSOC type while being lower in mucinous and early-stage tumors [109]. 

#### 3.1.5. TTR

Transthyretin (TTR) is a protein that transports thyroxine and retinol to the liver. Contrary to the previously discussed markers, its serum concentration is decreased in ovarian cancer. TTR performed better than CA125 and HE4 in detecting early-stage ovarian cancer; however, because of low sensitivity, it proved more efficient when combined with other biomarkers [110]. 

#### 3.1.6. Multivariate Index Assays

Further need to improve diagnostic accuracy brought about the implementation of multivariate index assays. One of the first such approaches was the introduction of serial CA125 measurements and their use in calculating the ROCA (Risk of Ovarian Cancer Algorithm). The ROCA used with transvaginal ultrasound (TVUS) in patients with increased serial CA125 levels improved sensitivity to 85% for earlier detection [111]. The UK Collaborative Trial of Ovarian Cancer Screening (UKCTOCS) revealed that longitudinal CA125 testing combined with annual TVUS screening improved the recognition capability for early-stage ovarian cancer by 47%. Still, it was not accompanied by a significant decrease in mortality [112].

The Risk of Ovarian Malignancy Index (ROMI) is a multivariate index assay based on serum levels of CA125, HE4, and thymidine kinase-1 (TK1). It showed higher specificity and sensitivity than ROMA in premenopausal and postmenopausal women [113]. 

The subsequent multivariate index assay is OVA1, which combines serum biomarkers CA125, transthyretin, transferrin, beta-2 microglobulin, and apolipoprotein A-1 to calculate a risk score for ovarian cancer. Compared to CA125 alone, OVA1 performs better concerning the sensitivity and identification of 60–80% of early-stage ovarian cancer cases missed by CA125 serum measurements [114,115]. OVERA was created as the second-generation test for OVA1 and combined CA125, HE4, apolipoprotein A1, transferrin, and follicle-stimulating hormone for screening patients with a pelvic mass. Compared to OVA1, OVERA had improved sensitivity and specificity of 91% and 69%, respectively [116]. 

Combining transthyretin (TTR) with CA125, hemoglobin, apolipoprotein AI, and transferrin improved the detection of early-stage ovarian cancer compared to CA125 [117]. Another combination of CA125, HE4, leptin, osteopontin, and prolactin showed improved AUC compared to CA125 alone [118]. CA125, in combination with annexin A2 (ANXA2), showed 100% sensitivity and improved accuracy in distinguishing stage IA ovarian cancer from benign tumors [119]. The following two multivariate models not only performed better but also anticipated the diagnosis of ovarian cancer by 5 months to 2 years compared to CA125 alone. The first model used insulin-like growth factor-binding protein-2 (IGFBP-2), lecithin cholesterol acyltransferase (LCAT), and CA125. In contrast, the second one was based on a combination of CA125, HE4, arginase 2 (AGR2), phosphatidylethanolamine binding protein 4 (PEBP4), and chitinase-3-like protein 1 (CHI3L1) [120,121]. A combination of ROMA, HE4, CA125, the neutrophil-to-lymphocyte ratio (NLR), and lactate dehydrogenase (LD) with machine-learning algorithms could improve the prediction of ovarian cancer with AUC from 0.62 to 0.76. Both LD and NLR were correlated with survival in ovarian cancer patients [122]. In the European Prospective Investigation into Cancer and Nutrition cohort of patients and ovarian cancer-free women, nine possible markers were chosen from 92 preselected proteins. These markers included CA125, HE4, FOLR1, kallikrein-related peptidase 11 (KLK11), WNT1-inducible-signaling pathway protein 1 (WISP1), midline or neurite growth-promoting factor 2 (MDK), C-X-C motif chemokine ligand 13 (CXCL13), mesothelin (MSLN), and ADAM metallopeptidase domain 8 (ADAM8). These markers showed good discrimination between groups for a lag time of 0–9 months; however, none of the new markers improved the prediction when added to CA125 [123]. The panel of 26 proteins in plasma associated with the dysregulation of Hippo, Hippo-Merlin, Focal-Adhesion-PI3K/AKT/mTOR, PI3K/AKT, Ras, Ebola virus, and NOTCH signaling pathways was able to identify patients with HGSOC risk at a false discovery rate of <0.05 [124].

#### 3.1.7. Urinary Tests

The evaluation of urinary biomarkers for the detection of ovarian cancer is a new and dynamic diagnostic possibility. Similar to serum-based studies, urine component searches are based on single or multiple markers. The marking of CA125 in urine samples gave a sensitivity of 89% and a specificity of 67% [125]. The subsequent urine marker tested was HE4, which provided a sensitivity of 51–89% and a specificity of 75–100%, depending on the study, and showed comparable accuracy to the serum testing [126,127,128]. The mesothelin urine assay showed that a more significant fraction of patients with both early-stage and advanced-stage ovarian cancer was detected with a urine assay (42% and 75%, respectively) compared to a serum assay (12% and 48%, respectively) [129]. Elevated levels of the anti-apoptotic protein Bcl-2 were present in urine from patients with ovarian cancer, independently from tumor stage, grade, and size, and differentiated between healthy and ovarian cancer patients with an AUC of 0.9 [130]. Urine high mobility group AT-hook 1 (HMGA1) protein was significantly elevated in serous epithelial ovarian cancer, and a higher AUC (0.88) was noted for urine HMGA1 than for serum CA125 [131]. The AUC of 0.68 for differentiating ovarian cancer from benign disease was pointed out for the urine minichromosome maintenance complex component 5 (MCM5) protein concentration [132]. 

Similarly to the serum, the multiple markers were also marked in urine. The combination of HE4, creatinine, carcinoembryonic antigen (CEA), and transthyretin (TTR) had 93.7% sensitivity and 70.6% specificity in predicting ovarian cancer compared to benign tumors [133]. The different combinations of the urine levels of two metalloproteinases (MMP-2 and MMP-9) and MMP-9/neutrophil gelatinase-associated lipocalin (NGAL) complex and age were evaluated in ovarian cancer compared to benign tumors. The best prediction with an AUC of 0.88 was obtained for MMP-2, MMP-9, and age [134]. Combining urinary eosinophil-derived neurotoxin (EDN) and osteopontin resulted in a sensitivity of 72% and specificity of 95% to distinguish ovarian cancer from normal controls and elevation in early-stage ovarian cancer [135]. MiR-30a-5p was shown to be up-regulated in HGSOC, while urinary miR-30a-5p was notably reduced following the surgical removal of ovarian tumors. The miR-30a-5p was concentrated within the exosomes from ovarian cancer or urine from HGSOC patients. MiR-30a-5p showed good discrimination between HGSOC patients and healthy controls with an AUC of 0.86 [136]. Several proteins, including LY6/PLAUR domain containing 1 (LYPD1), lymphatic vessel endothelial hyaluronan receptor 1 (LYVE1), prothymosin alpha (PTMA), and secretoglobin family 1A member 1 (SCGB1A1) were confirmed to be increased in the urine of ovarian cancer patients compared to benign ovarian tumors, giving an AUC of 0.92 [137]. The combination of fibrinogen α fragment, collagen α 1 fragment, and fibrinogen β NT fragment showed an AUC of 0.88 for discrimination between ovarian cancer and benign ovarian tumors. The AUC was enhanced to 0.96 by an additional combination with CA125 [138].

#### 3.1.8. Epigenetic Markers

Micro RNAs belonging to non-coding regulatory RNAs play an essential role in changing the expression of many target genes and intercellular communication. The levels of several miRNAs were investigated in serum, plasma, and circulating peripheral blood exosomes. The panel of eight miRNAs was both up-regulated (miR-21, miR29a, miR92, miR-93, and miR-126) and down-regulated (miR-99, miR-127, miR-155) in the serum of ovarian cancer patients compared to normal controls [139]. In the next studied panel of plasma 11 miRNAs, the up-regulation of miR-191-5p, miR-206, miR-34c-5p, miR-548a-3p, miR-320a, miR-574-3p, miR-590-5p, and miR-106b-5p and the down-regulation of miR-19a-3p, miR-30a-5p, miR-645, and miR-150-5p was noticed [140]. Similarly, serum expression of miR-145 and miR-133b was significantly decreased in ovarian cancer patients [141]. Moreover, the combined detection of miR-193-5p with CA125 and HE4 improved the accuracy of the diagnosis of ovarian cancer with an AUC of 0.996 [142]. MiR-200a, miR-200b, and miR-200c were significantly increased in the serum of HGSOC patients, with miR0-200b + miR-200c having the power to discriminate between ovarian cancer and healthy patients with an AUC of 0.78 [143]. A set of ten different miRNAs (miR-4687-3p, miR-939-5p, miR-5739, miR-211-3p, miR-1273g-3p, miR-3663-3p, miR-4726-5p, miR-4745-5p, miR-1268b, and miR-658) provided a prediction of ovarian cancer with an AUC of 0.97. However, every individual miRNA from the set was insufficient to discriminate cancer patients from healthy subjects when used alone [144]. The different sets of ten serum miRNAs enabled the identification of patients at risk of HGSOC. The set was constructed from miR-5100, miR-6800-5p, miR-1233-5p, miR-4532, miR-4783-3p, miR-4787-3p, miR-1228-5p, miR-1290, miR-3184-5p, and miR-320b and identified at-risk patients with an AUC of 1.00 [145]. A serum miRNA neural network of seven miRNAs (miR-29a-3p, miR-92a-3p, miR-200c-3p, miR- 320c, miR-335–5p, miR-450b-5p, and miR-1307–5 p) tested on an independent external sample of ovarian cancer showed a diagnostic AUC of 0.85. Moreover, the in situ hybridization of three miRNAs from the network (mir-200c-3p, mir-335–5 p, and mir-92a-3p) showed a complete overlap between the serum miRNAs and the miRNAs expressed in serous tubal intraepithelial cancer (STIC) and early stage I HGSOC [146]. The combination of miR-320b and miR-141-3p alone or integrated with CA125 and HE4 showed the accurate discrimination capability of early-stage ovarian cancer and healthy controls (AUC 0.79 and 1.00, respectively) [147]. Finally, miR-204, CA125, and CA19.9 panels showed an AUC of 0.94, 1.00, and 0.99 for benign, early-stage, and advanced ovarian cancer tumors, respectively [148]. Besides miRNA, another group of non-coding RNA called circular RNA has been tested for diagnostic accuracy. Circ _0003972 and circ_0007288 were down-regulated in the plasma of ovarian cancer patients and acted as diagnostic markers with an AUC of 0.78 and an AUC of 0.92 when combined with CA125. Both circ_RNAs function as tumor suppressors by sponging oncogenic miRNAs [149]. Tumor-associated molecules could be transferred to platelets, which are then educated to induce the specific splicing of pre-messenger RNA. These are called tumor-educated platelets (TEPs) and were tested as diagnostic tools in differentiating between early-stage ovarian cancer and benign tumors with 80% accuracy [150]. The usefulness of TEPs is evaluated in clinical trials (ClinicalTrials.gov identifier: NCT04022863). 

An exciting possibility comes from the study demonstrating the use of a DNA-methylation signature of cervical cells to identify both ovarian and endometrial cancer. The signature Women’s Risk Identification for Ovarian Cancer Index (WID-OC index) enabled the identification of women with ovarian cancer with an AUC of 0.76. Interestingly, the WID-OC index was not connected with the presence of ovarian tumor DNA in cervical samples but rather with cervical epigenetic profiles originating from disturbed Müllerian duct differentiation [151]. Testing the DNA methylation according to meta-analysis showed better performance for the prediction of ovarian cancer than cell-free tumor DNA with an AUC of 0.93 [152]. The homeobox A9 (HOXA9) and hypermethylated cancer-1 (HIC1) promoter methylation in serum cell-free DNA samples showed an AUC of 0.95 in discrimination between ovarian cancer and healthy controls [153,154]. The plasma cell-free DNA (cfDNA) methylation pattern has been shown to have diagnostic potential. The detection model based on cfDNA methylation achieved a sensitivity of 95% and a specificity of 89%, outperforming the CA125 [155]. 

#### 3.1.9. Autoantibodies (AAbs)

The most common mutation in HGSOC is the *TP53* mutation, which is present in over 95% of cancer cases [156]. The mutation is observed even in the early stages of HGSOC, and the lead time to the diagnosis using anti-TP53 serum AAbs is about 8–9 months, enabling the earlier prediction of cancer than with the use of CA125 or ROCA [157]. However, the anti-TP53 indicated minimal sensitivity as a single marker. Moreover, the presence of anti-TP53 AAbs is observed in only 40% of HGSOC patients and correlates with tumor burden. The combination of anti-TP53 AAbs with CA125 improved the accuracy of CA125 alone. The panel of three anti-tumor-associated antibodies, anti-G-protein alpha subunit Gs-α (GNAS), anti-TP53, and anti-nucleophosmin 1 (NMP1), could identify 51% of ovarian cancer patients negative for CA125 [158]. Paraneoplastic tripartite motif-containing protein 21 (TRIM21) AAbs in combination with anti-New York esophageal squamous cell carcinoma 1 (NY-ESO-1), anti-TP53, and anti-paired box 8 (PAX8, marker of the Fallopian tube secretory cell lineage) AAbs tested in the group of patients with increased risk of ovarian cancer (positive family history, Lynch syndrome, or *BRCA* germinal mutations) showed 46% sensitivity and 98% specificity. It could complement CA125 screening in this group of women [159]. Another panel of nine AAbs (anti-TP53, C-myc, protein kinase p90, ubiquitin-binding p62, alpha 2-HS glycoprotein AHSG, ubiquitously expressed 14-3-3zeta, Ras-related protein RalA, KH domain-containing protein over-expressed in cancer Koc, and tumor suppressor p16) showed an AUC of 0.91 and identified 79% of ovarian cancer patients presenting with normal CA125 levels [160]. Autoantibodies against leucine repeat death domain-containing protein (LRDD), stanniocalcin-1 (STC1), and forkhead box a1 (FOXA1) were over-expressed in ovarian cancer patients compared to normal controls and showed an AUC of 0.91, 0.88 and 0.82, respectively. The best discriminating combination was anti-LRDD + anti-FOXA1 AAbs, and the accuracy of this set improved when further combined with CA125 [161]. This promising combination was based on anti-BRCA1-associated RING domain 1 (BARD1) and CA125 levels. It showed high specificity and sensitivity (over 90%) for distinguishing ovarian cancer patients from healthy controls both for average-risk and high-risk women with hereditary breast and ovarian cancer syndrome (HBOC) [162]. Anti-PDZ and LIM domain-1 (PDLIM1) AAbs were able to differentiate both ovarian cancer patients from healthy controls (AUC 0.76) and ovarian benign tumors (AUC 0.76), as well as to identify 41% of early-stage and 39% of late-stage ovarian cancer, respectively. The anti-PDLIM1 AAbs could also find 15% of ovarian cancer patients negative on the base of CA125 marking alone. The combination with CA125 improved the AUC to 0.85 [163]. The panel of serum C-C motif chemokine ligand 18 (CCL18) and C-X-C motif chemokine ligand 1 (CXCL1) antigens combined with nucleic acid-binding protein (C1D), fragile X mental retardation syndrome-related protein 1 (FXR1), zinc finger protein 573 (ZNF573) and transmembrane 4 L six family member 1 (TM4SF1) AAbs showed an AUC of 0.99 for ovarian cancer diagnosis, an AUC of 0.95 for early-stage ovarian cancer, and an AUC of 0.75–0.96 for discriminating patients with ovarian cancer from patients with other malignancies [164]. The systematic review from Heidelberg [165] showed that the highest individual sensitivity was obtained with anti-Rho GDP-dissociation inhibitor (RhoGDI) and anti-tubulin alpha 1c (TUBA1C) AAbs but with concomitant low specificity. Better accuracy was obtained for anti-homeobox A7 (HOXA7) and anti-interleukin-8 (IL8) AAbs for detecting moderately differentiated and early-stage tumors, respectively. The panel of eleven AAbs (intercellular adhesion molecule 3-ICAM3, cancer/testis antigen 2-CTAG2, transcription factor p53, serine/threonine/tyrosine interacting like 1-STYXL1, PVR cell adhesion molecule-PVR, proopiomelanocortin-POMC, Nudix hydrolase 11-NUDT11, tripartite motif containing 39-TRIM39, serine/threonine protein kinase Kist-UHMK1, kinase suppressor of ras 1-KSR1, and nuclear RNA export factor 3-NXF3) provided 45% sensitivity at 98% specificity for HGSOC. Generally, limited data suggest that AAbs could improve diagnostic discrimination when combined with CA125 and HE4 [165].

### 3.2. Assessment of Adnexal Mass-Prediction of Ovarian Cancer in Symptomatic Patients

#### 3.2.1. IOTA-SR

The IOTA Simple Rules were devised by the International Ovarian Tumor Analysis (IOTA) group to predict malignancy of the recognized adnexal mass [166]. The model is based on the ultrasound estimation of B (benign) and M (malignant) features of the adnexal tumor, which helps to classify the patient into gynecologic oncology assessment. IOTA-SR predicts ovarian malignancy with a 92% sensitivity and 96% specificity and is more accurate than RMI or ROMA alone [167]. Therefore, the IOTA should be the first method to assess a pelvic mass, and its results should be confirmed using an expert ultrasound. We believe the IOTA system could be used for screening in postmenopausal women. 

#### 3.2.2. Multivariate Assays

The Risk of Malignancy Index (RMI) combines TVUS, CA125 level, and menopausal status and expresses the result as a calculated product of these components. The cut-off value of 200 has better sensitivity and specificity than the CA125 level alone [168,169]. Using different values for TVUS and menopausal status, four RMI algorithms were created; however, their clinical efficacy illustrated by the AUC did not prove to differ between them [170]. 

The multivariate index Risk of Malignancy Algorithm (ROMA) has incorporated the CA125 serum level, HE4, and menopausal status [111]. Compared to other assays, ROMA showed satisfactory specificity (91%) and sensitivity (90%) in the diagnosis of ovarian cancer [171]. However, the ROMA index showed a lower specificity in postmenopausal compared to premenopausal women [172]. Compared to CA125, ROMA showed improved specificity, especially in premenopausal women, as well as higher sensitivity compared with the RMI [173].

### 3.3. Screening Trials

The problem of screening for ovarian cancer has not been solved so far, although several attempts at screening in the average-risk postmenopausal women population have been undertaken. The prevalence of pathology in the population determines the demands for effective screening, which, in the case of ovarian cancer, taking into account the incidence of 40 cases per 100,000 persons/year, means that the test accuracy required to achieve a 10% positive predictive value in the whole population demands a sensitivity of 60–100% and a specificity of 99.6–99.8% [174]. In postmenopausal women, the test should have a sensitivity more significant than 75% and a specificity greater than 99.6% to reach a positive predictive value of 10% for the detection of early cancer. That means that based on the screening test itself, it would be nine false positive cases per one case of true ovarian cancer, which is an acceptable result. Using a single test is insufficient to attain this screening efficacy level. Therefore, the solution is to use either multimodal testing or to combine it with serial testing [175]. Several attempts at screening in the average-risk postmenopausal women population have been undertaken. In the Kentucky Study, women over 50 with no risk and women over 25 from risk families were included and received annual transvaginal ultrasounds, followed by detailed diagnostic procedures in suspected cases. The 5-year survival rate for women with ovarian cancer detected by screening was 75% compared with 54% for unscreened women with ovarian cancer treated in the same way [176]. The Shizuoka Cohort Study of Ovarian Cancer Screening (SCSOCS) recruited asymptomatic postmenopausal women for annual transvaginal ultrasound and CA125 (intervention group *n* = 41,688 vs. control group *n* = 40,799). Eight more cancers were diagnosed outside the screening program. The proportion of early-stage ovarian cancer was higher in the screened group (63%) than in the control group (38%); however, this did not reach statistical significance [177]. Extended follow-up of the Prostate, Lung, Colorectal, and Ovarian (PLCO) Cancer Screening Trial indicated no mortality benefit from screening for ovarian cancer with annual CA125 and transvaginal ultrasound [178]. However, the UK Collaborative Trial of Ovarian Cancer Screening (UKCTOCS) showed an overall mortality reduction of 20% in favor of multimodal screening (ROCA algorithm + transvaginal ultrasound in the group of patients with raised ROCA) [168]. In the subsequent trial, called the Normal Risk of Ovarian Cancer screening study (NROSS), postmenopausal women were tested using the ROCA algorithm and then selected for consecutive annual CA125 tests (low risk), repeat CA125 test in 3 months (intermediate risk), or transvaginal ultrasound and gynecologic oncologist consultation (high risk). The authors concluded that ROCA followed by transvaginal ultrasound demonstrated 99.9% specificity in a population of women at average risk for ovarian cancer [179]. Due to these discrepancies, screening programs have yet to be implemented commonly in clinics, and the efficacy of such proceedings still needs confirmation in randomized trials. An interesting problem is the question of how early the increase in biomarkers precedes the diagnosis of ovarian cancer (pre-diagnostic efficacy). Generally, for CA125, the lead time (the time to diagnosis) is shorter than one year and, in most studies, does not overrun six months. It has a higher sensitivity for advanced cancer (FIGO III-IV). The same is true for other biomarkers (HE4, CA72.4); however, panels of biomarkers outperform single biomarkers like CA125, having a longer lead time [120,180].

Another problem is screening in the population of women at risk of ovarian cancer. Meta-analysis of risk reduction estimates associated with risk-reducing salpingo-oophorectomy (RRSO) in *BRCA1* or *BRCA2* mutation carriers indicated that RRSO was associated with a statistically significant reduction in the risk of *BRCA1/2*-associated ovarian or fallopian tube cancer (HR 0.21) [181]. An NRG Oncology/Gynecologic Oncology Group study was performed on the GOG-0199 cohort of women at risk for ovarian cancer (carriers of *BRCA1/2* mutations or women with positive ovarian cancer history in first- and second-degree relatives) compared the RRSO and ovarian cancer screening (OCS) based on the ROCA algorithm and transvaginal ultrasound. Women choosing RRSO were older, found RRSO to be an effective procedure, and worried about OCS limitations. Women choosing OCS were more concerned about menopausal symptoms, infertility, and “loss of femininity” and reported better quality of life [182]. Phase II of the United Kingdom Familial Ovarian Cancer Screening Study (UKFOCSS) enrolled women with an elevated risk of ovarian/fallopian tube cancer. It subjected them to almost five years of observation using ROCA screening supplemented with annual transvaginal ultrasound. The screening resulted in a stage shift toward earlier diagnosis (53% at stage I/II); however, its influence on overall survival is unknown [183]. Despite these uncertainties, the National Comprehensive Cancer Network (NCCN) allowed screening with combined CA125 and transvaginal ultrasound in high-risk women aged 30–35 who did not want to undergo RRSO [184]. 

The accuracy of biomarkers and information about the studied groups is summarized in Table 2.

### 3.4. Liquid Biopsy

#### 3.4.1. Circulating Tumor Cells

Circulating tumor cells (CTCs) are the cells shed from the tumor, intravasated, and found in the peripheral blood. The isolation of CTCs is challenging due to the relative scarcity of tumor cells in all circulating cells, mainly of hematological lineage [201]. Most CTCs undergo both apoptosis and necrosis due to the hostile environment in the blood, including starvation, stress, and immunologically mediated elimination [202]. The most popular and approved system to isolate CTCs is a CellSearch detection system based on isolating cells with positive epithelial cell adhesion molecule (EpCAM) expression [203]. However, its efficacy is limited due to varied numbers of EpCAM(+) CTCs [204]. Novel approaches capable of omitting this limitation use CTC enrichment methods with multiple identification markers or the utilization of CTC’s ability to invade the artificial matrix [205,206,207]. 

One of the first publications devoted to CTCs identified them in the blood of 12% of ovarian cancer patients, using an immuno-microbead that recognized an EpCAM, regularly expressed on ovarian cancer cells [208]. Another early paper confirmed the evidence of CTCs in the peripheral blood of 19% of ovarian cancer patients, especially with most malignant grade 3 tumors. The CTCs were identified using cytokeratins CK7, CK8 CK18, and CK20, transcription factor-2 (TFS-2), and epithelial growth factor receptor (EGFR) [209]. The study detecting CTCs identified as cells positive for EpCAM, CK 4, 5, 6, 8, 10, 13, and 18 showed CTC positivity in 10% of early-stage cancer patients and 73% of late-stage patients. Patients with advanced cancer showed higher numbers of peripheral blood CTCs [199]. In another study, CTCs were detected before surgery in 19% of patients expressing EpCAM (31%), MUC-1 (50%), HER-2 (31%), and CA 125 (50%), respectively [210]. In another study, CTCs were isolated on the base of EpCAM/CK positivity combined with physical cellular features like cell size and deformability in patients with benign and malignant ovarian tumors. A total of 56% of cases were positive for CTCs, including 100% early-stage and 68% advanced-stage cancer. However, 44% of patients with benign tumors also showed the presence of cells interpreted as CTCs. The CTC detection had a sensitivity of 77%, 100%, and 100% for benign vs. all-stage cancer, benign vs. stage I-II cancer, and benign vs. stage I cancer, respectively [195]. In the next study, the CTC count established by a microfluidic method differed between healthy and ovarian cancer patients. CTCs defined as Hoechst+, CK+, and CD45- cells showed counts of more than eight in 87% of ovarian cancer patients but in none of the healthy volunteers, and the total CTC counts were found to be significantly elevated in the ovarian cancer group (55 vs. 0.5) even though CTC-like cells were identified in low numbers in healthy controls [211]. The immunomagnetic targeting of combined mesenchymal N-cadherin and epithelial EpCAM-positive cells enriched the CTC population approximately three times more efficiently than targeting EpCAM alone. According to previous studies, in some blood samples of healthy individuals, the presence of cells expressing markers common to CTCs was also observed. Analysis showed that these “false positives” were identified as circulating endothelial cells (CECs) by vascular endothelial-cadherin co-staining and that their count could be highly variable in patients and healthy controls [212]. This is a reliable explanation for the identification of CTC-like cells in benign tumors and healthy individuals. The CTCs marked as CD45-/HE4+EpCAM+cytokeratin+vimentin+ cells were increased in ovarian cancer patients and showed positivity in 49% of patients with a sensitivity of 73%, higher than that for CA125 [190]. Comparison of the gene expression level in the group of CTC-negative and -positive peripheral blood samples confirmed a statistically significant difference for the expression of the Wilms tumor *WT1*, *EPCAM*, *MUC16*, *MUC1*, keratin *KRT7*, *KRT18*, and *KRT19* genes [213]. Another study of the genetic pattern of epithelial EpCAM+ CTCs showed the expression of *MUC1* and cytokeratin 19 (*CK19*) but also of genes associated with mesenchymal and more malignant features such as tissue inhibitor of metallopeptidase 1 (*TIMP1*), C-X-C chemokine receptor type 4 (*CXCR4*), and the stem markers *CD24* and *CD44* [214]. 

The evolution of isolation techniques improved the CTC detection rate to almost 90% of ovarian cancer patients and increased the number of CTCs correlated with the clinical stage of the tumor. Moreover, patients with early-stage disease had a CTC-positive rate of 93% compared to CA125, which was positive at merely 64% [215]. The combination of anti-EpCAM-moAbs and anti-FRα-moAbs showed a significantly increased positive rate of CTC detection in ovarian cancer patients compared with anti-EpCAM-moAbs alone [216]. Recently, the detection model of CTCs based on the expression of EpCAM, MUC1, and Wilms tumor protein WT1 showed significantly higher specificity than CA125 (92% vs. 82%), especially in early-stage ovarian cancer (74% vs. 58%). The sensitivity of the detection, as mentioned in the above model, ranged up to 79.4% [196]. 

#### 3.4.2. Cell-Free DNA/Circulating Tumor DNA

Cell-free DNA (cfDNA) is a DNA released from apoptotic or necrotic cells and contains a subpopulation of circulating tumor DNA (ctDNA) in the case of cancer. The ctDNA is a more achievable target than CTCs due to its relative abundance [217]. The amount of cfDNA varies in plasma in wide borders (3–93%) and depends on the cancer’s presence and the tumor size [218]. The ctDNA fragments’ length is usually shorter than cfDNA; therefore, searching for shorter fragments could enrich the sample in ctDNA [219]. Moreover, cfDNA Integrity (cfDI) indicates the ratio of long necrosis-derived DNA fragments to short apoptosis-derived fragments. Cancer patients have a higher cfDI compared to healthy controls and benign tumors [220]. The stability of cfDNA in the blood depends on the amount of DNA released and the clearance of DNA, which, in the case of cancer, is significantly decreased [221]. The estimated time of the half-life of cfDNA was around 4–30 min [222]. The ctDNA detection is based on several methods, including quantitative polymerase chain reaction (PCR), digital droplet PCR, next-generation sequencing (NGS), and whole-genome sequencing (WGS) [223]. Combining NGS with targeted error correction sequencing (TEC-Seq) or duplex sequencing can improve further detection accuracy [224,225].

The main challenge in ovarian cancer is to recognize tumors in their early stages when surgery is much easier and the prognosis is better than in the advanced stage. The early phenomenon in ovarian cancer is promoter methylation of suppressor genes, which could be used as a diagnostic tool [226,227]. The methylation of opioid-binding protein/cell adhesion molecules like *OPCML*, Runt-related transcription factor 3 (*RUNX3*), and tissue factor pathway inhibitor 2 (*TFPI2*) genes was studied in ovarian cancer patients using methylation-specific NGS. The methylation of *OPCML* showed a significant difference between early-stage ovarian cancer patients and healthy controls, even when CA125 did not differ between them. Ras-association domain family 2A (*RASSF2A*) is a suppressor gene whose down-regulated expression has been described in several malignancies. The epigenetic inactivation of *RASSF2A* through aberrant promoter methylation was found in 36% of ovarian cancer plasma samples [228]. The multiplex assay of aberrantly methylated seven genes (*RASSF1A*, *RUNX3*, *TFPI2*, *OPCML*, secreted frizzled-related protein 5-*SFRP5*, cadherin 1-*CDH1*, and APC regulator of WNT signaling pathway-*APC*) achieved a sensitivity of 85% and a specificity of 90% in early-stage ovarian cancer, which was surprisingly better than the result obtained for CA125 [200]. The following suppressor gene, the slit homolog 2 (*SLIT2*), was shown to be aberrantly methylated. Among the cases with hypermethylation in ovarian tissue, 93% of the case-matched serum DNA samples also showed hypermethylation, including cases of early-stage ovarian cancer [229]. The methylation of the three genes *RASSF1A*, calcitonin-related polypeptide alpha (*CALCA*), and adenovirus E1A-associated cellular p300 transcriptional co-activator (*EP300*) enabled differentiation between ovarian cancer patients and healthy controls with a sensitivity of 90% and a specificity of 87%, while the methylation of *RASSF1A* and progesterone receptor-Prospero homeobox protein 1 (*PGR-PROX*) discriminated between ovarian cancer and benign tumors with a sensitivity of 80% and a specificity of 73% [197]. The methylation pattern of seven genes, frizzled receptor proteins 1, 2, 4, 5 (*SFRP1*, *2*, *4*, *5*), SRY-box 1 (*SOX1*), paired box gene 1 (*PAX1*), and LIM homeobox transcription factor 1 alpha (*LMX1A*), revealed the significant correlation between tumor and serum samples and high sensitivity and specificity (73% and 75%) as a screening marker [230]. Again, the methylation profiles of the *RASSF1*, *CDH1*, *PAX1*, and PTEN phosphatase and tensin homolog (*PTEN)* tumor suppressor genes were analyzed, and higher plasma values of methylation were found for *CDH1* and *PAX1* genes in malignant ovarian lesions [231]. Serum promoter methylation of homeobox A9 (*HOXA9*) and hypermethylation in cancer 1 (*HIC1*) showed a combined AUC of 0.95, and no hypermethylation was observed in sera from matched cancer-free control women [154]. The methylation pattern of serum 56 genes in HGSOC showed 85% sensitivity and 61% specificity in differentiating between HGSOC and healthy volunteers. Although the accuracy of this test was not optimal, one can expect that a proper choice of target genes will improve it to a clinically relevant level [198]. The pattern of cfDNA methylation discriminated HGSOC patients from healthy women or patients with a benign pelvic mass with 91% specificity and 41% sensitivity and had the potential to detect a proportion of ovarian cancers up to two years before the diagnosis [232]. 

Another possibility is to identify mutations in target genes by analyzing ctDNA fragments. CancerSEEK, a non-invasive blood test detecting ctDNA, was used to diagnose eight cancers, including ovarian cancer, and performed with a sensitivity of 69–98% with specificity greater than 99%. It also restricted the localization of the tumor to a small number of anatomic sites in 83% of patients [233]. Using the deep sequencing of ctDNA, it was possible to amplify the selected regions of cancer-related *TP53*, *EGFR*, *BRAF*, and *KRAS* genes. Mutations of *TP53* at allelic frequencies of 2–65% were found with sensitivity and specificity of more than 97% [234]. Chromosomal instability (CI) is a common feature in ovarian cancer. The patterns of CI were detected in HGSOC and improved malignancy detection (AUC 0.94) compared to CA125 (AUC 0.78) and RMI index (AUC 0.81) [191]. An exciting option both for clinical and commercial purposes is a multi-cancer early detection test (MCED), which was devised for the detection of multiple cancer types and tested in The Circulating Cell-free Genome Atlas (CCGA) prospective multi-center case-control study NCT02889978 accumulating 15,254 participants (56% with and 44% without cancer, 65 with ovarian cancer) [235]. The targeted methylation assay was conducted on cfDNA isolated from plasma samples. The study was validated on 3237 cancer and 2069 non-cancer patients. The overall specificity of the MCED test (Galleri TM) was 99.5%, while sensitivity was 51.5%. For ovarian cancer, sensitivity was estimated at 83.1% [235].

Low-coverage plasma DNA sequencing, usually used for non-invasive prenatal testing (NIPT), when used for the diagnosis of HGSOC, detected 41% of all HGSOC, including 38% of early-stage cases [236]. Chromosomal instability is a source of a disturbed copy number state, which is detectable in the ctDNA test. The copy number instability score (CNI-score) reached a sensitivity of 91% and a specificity of 95% to detect ovarian cancer in blood samples [237]. 

Beyond serum, other sources of ctDNA have been proposed. The PapSEEK test enabled an assay for mutations and aneuploidy of 18 genes. The sensitivity of Pap brush samples in ovarian cancer patients was 33%, including 34% of patients with early-stage disease. The sampling with an intrauterine brush improved sensitivity to 45%, and the combined use of a Pap brush and plasma ctDNA testing improved the sensitivity to 63% [238]. Lavage of the uterine cavity enables DNA material from cells shed from the ovarian tumor to fallopian tubes and uterus. The use of the NGS technique revealed the presence of mutations, mainly in *TP53*, in 60% of lavage samples of patients with ovarian cancer [239]. Similar results were obtained when *TP53* mutations were found in DNA extracted from vaginal tampons [240]. A nine-protein panel (myosin-11-MYH11, calcium-activated chloride channel regulator, 4-CLCA4, protein S100-A14, protein S100-A2, seeping B5, involucrin-IVL, CD109 antigen, nicotinamide N-methyltransferase-NNMT, ectonucleotide pyrophosphatase/phosphodiesterase family member 3-ENPP3) identified in exosomes obtained from uterine lavage was able to differentiate between HGSOC and healthy patients with 70% sensitivity and 76% specificity [241]. 

#### 3.4.3. Exosomes

Exosomes belong to the group of extracellular vesicles released from the cells. The other members of this group are microvesicles and apoptotic vesicles. Exosomes constitute two populations with diameters ranging from 60 to 80 nm for small exosomes and 90–160 nm for large exosomes and originate from the membranes of multivesicular bodies (MVB) [192,242]. Exosomes can carry different cargo and communicate with other cells in close neighborhoods and distant localizations. In cancer development, exosome-associated proteins, miRNAs, and metabolites could regulate cancer cells and stroma by autocrine or paracrine signaling. Endosome uptake by the target cells in the endocytosis mechanism enables the transferring of signals for tumor proliferation or the suppression of its growth [243].

In plasma exosomes of ovarian cancer patients compared to controls, the over-expression of fibrinogen alpha chain (*FGA*) and gelsolin (*GSN*) genes and the under-expression of fibrinogen gamma chain (*FGG*) and lipopolysaccharide-binding protein (*LBP*) genes, involved in coagulation and apoptosis, was noticed [193]. A panel of miRNAs found in exosomes isolated from plasma discriminated against ovarian cancer, benign tumors, and healthy women. MiR-21, miR-100, miR-200b, and miR-320 were significantly increased, whereas miR-16, miR-93, miR-126, and miR-223 were under-represented in exosomes from the plasma of ovarian cancer patients as compared to healthy women. The exosomal miR-23a and miR-92a levels were lower in benign ovarian cystadenoma patients than in ovarian cancer patients and healthy women, respectively. The exosomal levels of miR-200b correlated with the tumor marker CA125 and patient overall survival [194]. In another study, miR-93, miR-145, and miR-200c showed significantly higher expression in serum exosomes of the HGSOC group compared to the nonmalignant tumors group (borderline and benign tumors). MiR-145 showed the best AUC of 0.91 and a sensitivity of 91.6%, while miR-200c showed a specificity of 90.0%, better than CA125 [244]. The levels of eight exosomal miRNAs (miR-21, miR-141, miR-200a, miR-200c, miR-200b, miR-203, miR-205, and miR-214) isolated from the sera of ovarian cancer patients exhibited similar profiles different from miRNA profiles observed in benign ovarian tumors [245]. Another miRNA, miR-1290, was significantly over-expressed in the serum exosomes of malignant ovarian tumors compared to benign tumors (AUC 0.73) [246]. Exosomal levels of miR-99a-5p were up-regulated in the sera of ovarian cancer patients [247], as were other proteins, like L1 cell adhesion molecule (L1CAM), heat stable antigen CD24 (CD24), a disintegrin and metalloproteinase domain-containing protein 10 (ADAM10), extracellular matrix metalloproteinase inducer (EMMPRIN), claudin-4 (CLDN-4), and transforming growth factor beta (TGF-β) [248,249,250]. Uterine aspirates were shown to contain several both over-expressed (18 miRNAs) and under-expressed (11 miRNAs) exosomal miRNAs in ovarian cancer patients [251]. The exosome diagnostic platform ExoProfile chip combines eight markers (human epidermal growth factor receptor 2-HER2, epidermal growth factor receptor-EGFR, folate receptor alpha-FRα, CA125, EpCAM, CD24, a member of the transmembrane four superfamily CD9, and a protein associated with membranes of intracellular vesicles CD63) [252]. 

The usefulness of liquid biopsy for the diagnosis of ovarian cancer is presented in Table 3.

## 4. Biomarkers in Disease Monitoring

The CA125 biomarker was traditionally used to monitor patients with HGSOC after the first line of chemotherapy. However, even though increased CA125 blood levels may signal recurrent disease, the clinical value of early detection remains unclear, as it was studied in only one randomized controlled clinical trial [253]. A systematic review of studies describing the monitoring of CA125 increments in patients with advanced ovarian cancer during the follow-up period after the first-line chemotherapy showed that the median sensitivity was 85% (range 62–93%) [254]. False positive and false negative rates were 9% (0–33%) and 15% (7–38%), respectively. Most studies collected small and inhomogeneous patient populations. According to the Gynecological Cancer Intergroup (GCIG) criteria, progressive disease is defined as progression based on RECIST 1.1 or CA125 progression criteria [255]. The utility of GCIG criteria was estimated in several clinical randomized trials, and the conclusion was that CA125 increments could be used as an alternative to imaging techniques. It was of prognostic significance [256,257]. However, a meta-analysis of multiple studies indicated that high rates of false positive and false negative results could question the clinical utility of the CA125 criterion in clinical practice [254]. An analysis of CA125 criteria in a cohort of patients studied in phase III randomized trial AURELIA in platinum-resistant ovarian cancer enabled us to conclude that cancer progression was detected earlier by imaging than by the CA125 rise [258]. Moreover, neither overall survival nor quality of life improved due to therapy introduced based on the early CA125 rise [253]. The authors concluded that routine follow-up and information about the disturbing symptoms given to the patient should be offered instead of CA125 monitoring. At present, the recommendations for CA125 monitoring are very diverse. According to the European Society of Gynecologic Oncologists (ESGO) and the European Society of Medical Oncology (ESMO), abandoning CA125 monitoring is not advised as it may signal the tumor recurrence in symptomless patients or may be helpful in clinical trials in patients who refuse frequent follow-up and those eligible for secondary cytoreduction at recurrence [259,260].

Invasive CTCs were studied in flow cytometry assay, which showed 97% specificity and 83% sensitivity for detecting CTCs in patients’ blood and enabled the prediction of relapsed disease with higher sensitivity than CA125 (79.5% vs. 67.6%) [261]. The invasive CTC assay compared to CA125 alone also indicated a higher positive predictive value for the detection of early-stage ovarian cancer and the prediction of overall and progression-free survival [262]. Epithelial cell adhesion molecule EpCAM+/cytokeratin+, CD45-, and 4’,6-diamidino-2-phenylindole (DAPI)+ CTCs counts used for the serial monitoring of 13 ovarian cancer patients (10 with HGSOC and 11 with complete response to first-line chemotherapy) showed higher sensitivity (100.0% vs. 60.0%), positive predictive values (55.6% vs. 42.9%), and negative predictive values (100.0% vs. 87.5%) than CA125 levels. CTC counts were better associated with treatment response and recurrence than CA125 levels [263]. In patients during adjuvant chemotherapy, the numbers of CTCs increased significantly after surgery and decreased after the third cycle of chemotherapy. CTCs were detected in 100% of the chemotherapy-resistant patients but only in 91% of the chemosensitive patients. Moreover, the OS was significantly shorter in patients with EpCAM + CTCs than in patients with Ep-CAM-negative CTCs [215]. In the study group of 266 patients from the OVCAD consortium, baseline and follow-up CTC numbers were used to monitor the disease and to prognose patients’ outcomes. The CTCs were observed in 26.5% of baseline and 7.7% of follow-up blood samples. Baseline CTCs indicated a higher risk of death in patients with complete macroscopic cytoreduction, while at follow-up, the presence of CTCs was associated with poor response to chemotherapy [264]. It seems that chemotherapy preferentially segregates clonal tumor evolution toward therapy resistance, which was shown by the analysis of follow-up CTCs showing EMT markers. Their appearance after chemotherapy indicated a group of patients with decreased PFS and OS [265]. The excision repair cross-complementation group 1 (ERCC1) protein plays a role in nucleotide excision repair. The ERCC1-positive CTCs were detected in 12% of patients after chemotherapy and correlated with platinum resistance and reduced PFS and OS [266].

A few studies demonstrated that ctDNA concentrations correlate with responses to adjuvant treatment and may predict progression in advance compared to CA125 or imaging. Undetectable levels of ctDNA six months after the initial primary treatment were associated with significantly improved PFS and OS [267]. An analysis of TP53 mutations in the ctDNA showed that the number of detected gene copies was correlated with the volume of relapsed tumor, and a decrease of less than 60% in the TP53 mutant allele fraction after chemotherapy was associated with poor response and shorter time to progression [268]. Constantly detectable ctDNA levels after cytoreduction were consistent with residual disease and a recurrence risk [269]. The ctDNA analysis could be more sensitive and specific than the CA-125 levels when used for monitoring the response to therapy in advanced ovarian cancer patients, and more importantly, plasma ctDNA might be an independent factor of OS and PFS in patients with disease recurrence [270]. Aberrant methylation patterns detected in ovarian cancer cfDNA have been evaluated for patient monitoring in a single study. This study indicated that reduced methylated ctDNA levels in ovarian cancer patients after chemotherapy were associated with treatment response [191].

In summary, the usefulness of classic CA125 biomarkers in ovarian cancer monitoring has been doubted by studies showing a high rate of false results. The utility of CTCs and ctDNA for monitoring the disease still needs adequate estimation in randomized multicenter trials. The optimal isolation method of CTCs and candidates for ctDNA indicator genes are still disputable and need clinically helpful solutions. The positive observation is that both CTCs and ctDNA have great potential to become precise monitoring tools for ovarian cancer patients.

## 5. Future Directions

### 5.1. Risk Factors

The diagnosis of gene mutations in the relatives of ovarian cancer patients or in families with positive cancer history enables the identification of a population at risk, which could benefit from RRSO after the completion of procreation. This is realized today. The analysis of SNPs could improve the identification of patients with intermediate risk of HGSOC. Nowadays, it is still primarily a scientific problem, as practical implementation is complex due to cancer heterogeneity and unacceptable costs in the general population. However, identifying risk SNPs could broaden the population that should be subjected to RRSO. Therefore, finding an HGSOC-universal and economically approvable SNP-based test is still needed. 

Unfortunately, the studies devoted to the role of the microbiome in ovarian cancer do not indicate clearly if the microbiome changes are the ovarian cancer predisposing factor or a consequence of disturbed immunosurveillance and environment due to the progress of the cancer. The integration of the viral genetic material with the host genome weights is the first possibility; however, in the case of bacterial microbiota, the available data could speak for both options. The proper answer is even more problematic in the case of both fungal and parasitic microbiota. However, the attempts of microbiota exploration are worth continuing, as finding potentially causative or highly predisposing infectious factors could enable adequate prophylaxis (vaccination?) and treatment. Moreover, the immune status of cancer patients modulated by gut microbiota composition may change the response to anti-PD-1 immunotherapy [271,272]. The supplementation of probiotics could also improve the response to cancer therapy by reducing therapy-associated complications [273]. Finally, bacteria and their products could be used as vectors for drugs or toxic anti-cancer agents or as immune potentization of the patients [274]. As both genital tract and gut microbiota could influence the local environment in the pelvis, an attempt to change it by eradicating residing bacteria and replacing them with protective/typical strains could be of interventional relevance. An example of such intervention is fecal transplantation, which is approved for carriers of the Clostridium difficile infection, which is also being tested in other chronic colon diseases [275].

### 5.2. Prediction of HGSOC

The main obstacle is a lack of efficient and economically warranted screening programs for a population with an average risk of ovarian cancer. As a lifetime risk does not overrun 2% in this population, the money-consuming programs are not widespread. However, the cost of treatment in recognized ovarian cancer is so high and the psychological burden for cancer careers is so devastating that screening would be justified irrespective of economic challenge. The relatively cheap and still primary method is CA125 serial testing, supplemented by transvaginal ultrasound; however, CA125 does not cover the whole spectrum of epithelial ovarian cancer pathology due to low accuracy for early-stage HGSOC and borderline and mucinous cancers, as well as false positive results in some benign tumors. The multimodal combinations of serum markers could effectively support or replace CA125, and future efforts should concentrate on constructing such multimodal tests of highly acceptable specificity and sensitivity combined with low expenses for testing. Moreover, tests should also be available in developing countries where early diagnosis and sophisticated treatment methods are highly restricted. A very promising method of prediction and diagnosis of HGSOC is a liquid biopsy, using the identification of CTCs, ctDNA, or circulating non-coding RNAs. An attractive alternative is also seeking a unique genetic signature in material collected in a non-invasive manner; for instance, during liquid-based cytology or urine tests. The identification of families with a higher incidence of cancer, including ovarian tumors, performed by simple means of detailed family history collection in general practitioner or gynecologic clinics by way of routine visits, could enable the selection of women for genetic testing, including the identification of SNPs. This “at risk” group could be thoroughly monitored or subjected to prophylactic salpingo-oophorectomy during other surgical procedures. However, to achieve satisfactory accuracy, some obstacles should first be overcome. The detection rate of CTCs is not uniform and varies from 56–100%, being more reliable in advanced compared to early-stage ovarian cancer [205,208] due to the variability of isolation and evaluation methods. Moreover, CTC isolation needs to collect a relatively large amount of blood. A similar situation characterizes the ctDNA isolation. The detection rate varies from 38 to 100% and is usually greater in advanced cancer, which hampers the diagnostic potential of ctDNA testing in early-stage cancer [276]. In the case of miRNAs, the identification procedures are time-consuming and expensive, and some miRNAs are not specific for one tumor type, thus disabling the recognition of selective ovarian cancer [277]. MiRNAs in exosome cargo are more stable, but the isolation procedures are complicated and expensive; this should be one of the main future directions. The search for practical tests should be concentrated on the methods enabling the identification of early-stage cancer. 

The trend to personalize the treatment of ovarian cancer has been observed in recent years; however, the main achievements in this field that have hardly improved outcomes are only PD-1/PD-L1 and PARP inhibitors. Dozens of other drugs are being tested, but pre-clinical and clinical trials have given ambiguous results. In our opinion, the main obstacles in HGSOC treatment are its unique peritoneal/adipose tissue environment, the spatial and temporal heterogeneity (different therapy responses in primary/metastatic tumors and primary/recurrent tumors), and the vivid activity of cancer stem cells (sustained by a highly energetic, immuno-resistant “niche”). Therefore, anti-cancer management should take into consideration all these circumstances and should be fitted to the stage of the tumor, the time point in the course of the disease, and the patient’s general and immunological status. The basic rules of such a novel individualized approach to HGSOC therapy named the “DEPHENCE” system (“Dynamic PHarmacologic survEillaNCE”) have been proposed by us recently [278]. The basic assumption of the system is the personalization of the therapy, which should take into account the histotype, the molecular signature of ovarian cancer, and the stromal/immune composition of its microenvironment (i.e., represented by mesenchymal, differentiated, immunoreactive, and proliferative subtypes). The primary neoadjuvant or post-cytoreduction adjuvant treatment should be tailored to these subtypes. The therapy should attack the cancer cells, cancer stem cells, and tumor-supporting cells from their microenvironment, like cancer-associated fibroblasts, M2-type tumor-associated macrophages, and immune suppressor cells. As the tumor’s molecular signature differs between primary tumors, peritoneal implants/metastases, and recurrent tumors, the therapy should be appropriately adjusted to the particular phase and predominant localization of the disease. To do this, recurrent tumor biopsies should be obtained at every disease step (i.e., needle biopsy, laparoscopic biopsy, or liquid biopsy). In our opinion, the identification of the high-risk population of women and techniques of early detection based on the identification of cancer biomarkers should also be incorporated into the “DEPHENCE” system (Figure 1).

## Figures and Tables

**Figure 1 biomedicines-12-00229-f001:**
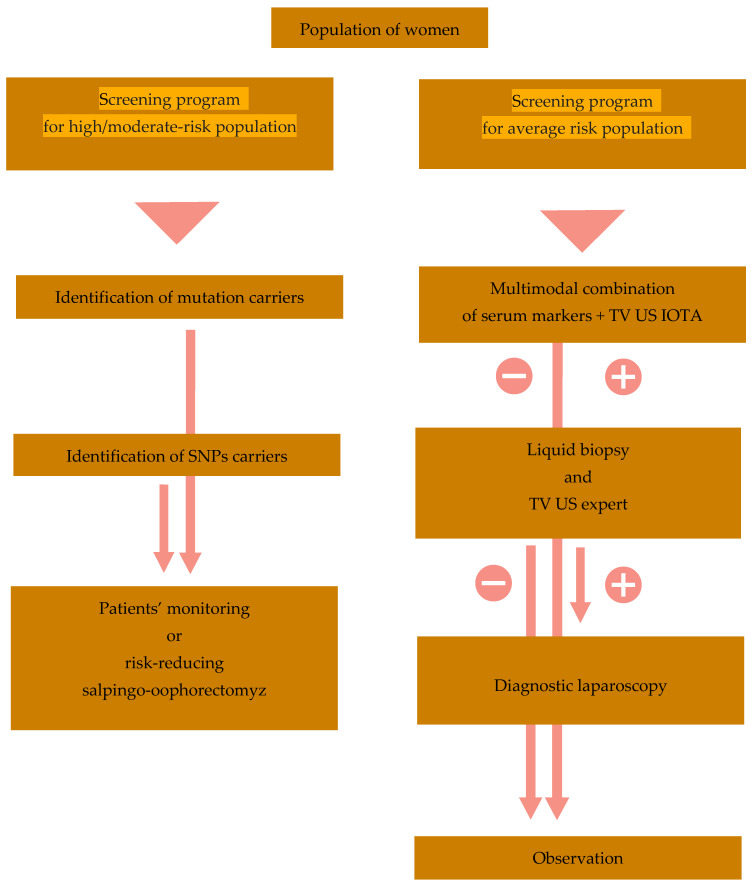
“DEPHENCE” system in ovarian cancer diagnosis. The general population of women should be divided into the population with an increased risk (identified based on the higher incidence of cancer in family members, the incidence of ovarian/breast cancer in first/second-degree relatives, and positive familial history of genetic syndromes with increased risk of cancer) and the population with an average risk of ovarian cancer. The higher-risk population should be subjected to the identification of gene mutations and/or SNPs, increasing the risk of ovarian cancer occurrence. In the case of positivity, the risk-reducing salpingo-oophorectomy or serial monitoring should be offered after completing reproduction. According to IOTA rules, the average-risk population should be tested with a serial multimodal combination of serum markers and serial transvaginal ultrasound. In the case of the positivity of either marker levels or the presence of a suspected adnexal tumor, the next step should be performing a liquid biopsy and expert ultrasound, and, when positive, eventual diagnostic laparoscopy. Patients with negative results of marker tests or ultrasound are directed to observation and further screening (once a year?). The multimodal combination of serum markers, as well as targets for the liquid biopsy, should be chosen on the basis of high specificity and sensitivity for early-stage ovarian cancer, as advanced tumors are easily recognizable. TV US—transvaginal ultrasound; IOTA—International Ovarian Tumor Analysis group; ctDNA—circulating tumor DNA.

**Table 1 biomedicines-12-00229-t001:** Epidemiological, genetic, and microbiota risk factors for HGSOC.

Risk Factor	Comment	Reference
	**Epidemiological factors**	
A higher number of ovulatory cycles during the lifetime	Disruption and subsequent healing of the ovarian epithelium during ovulation could account for ovarian cancer. Early menarche, late menopause, nulliparity, a lack of breastfeeding, and the non-usage of oral hormonal contraceptives are all factors influencing the number of ovulations.	[3,4,5,6]
Menopausal estrogen replacement therapy	The proliferative potential of estrogens for ovarian cancer development (especially endometroid histotypes) cannot be excluded.	[7]
Diabetes, obesity, cigarette smoking	All of them could account for a chronic inflammatory state in the tissues and the over-production of reactive oxygen species. Cigarette smoke contains several carcinogens.	[3]
A diet rich in animal proteins and fat	Contains xeno-estrogens, which have carcinogenic potential and stimulate leptin and gonadotropin secretion. It is followed by the disturbed proliferation of the ovarian epithelium after ovulation.	[9,10,11]
Tubal ligation, salpingo-oophorectomy, and hysterectomy are considered protective factors	Salpingo-oophorectomy removes both ovaries and tubes with fimbriae. Tubal ligation and hysterectomy break the connection between the lower genital tract and peritoneal cavity, which could disable the transmission of infective or irritative factors from the vaginal or external environment.	[5,8]
	**Gene mutations**	
Germline *BRCA1/2* mutation	Disruption of homologous recombination repair mechanism (homologous recombination deficiency-HRD). Germline *BRCA1/2* mutations are the most common cause of HRD.	[13,14,15]
Mutation of *BRIP1*, *RAD51C*, *RAD51D*, *PALB2*, *ATM*	Lower-penetrance genes responsible for HRD when mutated.	[16,17]
Lynch syndrome (hereditary nonpolyposis colorectal cancer syndrome-HNPCC)	Disturbed DNA mismatch repair system and microsatellite instability due to the *MLH1* and *MSH2* mutations.	[22]
Li–Fraumeni syndrome	An autosomal dominant syndrome determined by heterozygous germline mutations in the tumor suppressor gene *TP53*.	[26,27,28]
	**Gene polymorphism**	
*CHEK2* SNP rs17507066	A significant association with the risk of HGSOC. *CHEK2* is engaged in cell cycle arrest, DNA repair, and DNA damage-related apoptosis.	[20]
*PIK3CA* SNP rs9838117	Polymorphism could influence cell signaling, increasing the ovarian cancer risk.	[29]
*HEXIM1* SNP rs1053578	Correlated with an increased risk of ovarian cancer. *HEXIM1* functions as a regulator of gene expression.	[30]
*IGFBP3* SNP rs2270628	The *IGFBP3* gene functions as a low-penetrance onco-suppressor gene. Its polymorphism was associated with higher ovarian cancer risk.	[36]
*TLR4* SNP Asp299Gly	This SNP is a risk factor for the development and course of ovarian cancer. TLRs are engaged in the inflammatory pathways, and their activation in ovarian cancer is closely related to cancer aggressiveness, chemo-resistance, and adverse clinical outcomes.	[39,40,42]
*HLA-DP* rs3077 AA, *HLA-DQ* rs3920 GG, *ICAM-1* rs1437 CC, and CT genotypes	These genotypes are correlated with an increased risk of ovarian cancer and were found to be different in borderline versus malignant tumors.	[43]
*DNMT1* SNPs rs2228611 and rs759920	*DNMT* genes regulate the expression of DNA methyltransferases, which control DNA methylation, the process of epigenetic regulation of cancer growth.	[37]
miR-196a-2 rs11614913 CC genotype	It increases the risk of ovarian cancer and enhances cell invasion. Depending on the conditions, miR-196a-2 may function either as oncomiRs or as tumor suppressors.	[32]
*MSL1* SNP rs7211770	This SNP was associated with a decreased risk of HGSOC. The *MSL1* gene is engaged in the formation of higher-order chromatin structures.	[30]
*TCEAL7* SNPs rs5987515, rs5987724 and rs5945971	SNPs were significantly associated with a reduced risk of HGSOC. *TCEAL7* negatively regulates the NF-κB signaling and represses cyclin D1 expression.	[31]
*HEPH* SNP rs17216603	This SNP was associated with a decreased risk of HGSOC. *HEPH* regulates the proliferation and apoptosis of cancer cells and facilitates reactive oxygen species production.	[33]
*RB1* SNPs rs2854344 and rs4151620	SNPs were associated with a reduced risk of ovarian cancer. *RB1* is an onco-suppressor and regulates the cell cycle in cells with damaged DNA.	[32]
	**Microbiome**	
HPV 16 and 33 DNA is present in 10.5% of HGSOC samples CMV DNA was found in 40–50% of HGSOC samples HPV and CMV co-infection was observed in almost 60% of HGSOC samples, including fallopian tubes	HPV oncogenes *E6* and *E7* could stimulate the degradation of p53 and Rb proteins, which trigger carcinogenesis. CMV proteins can inhibit functions of p53 and modulate the function of many, thus stimulating proliferation, angiogenesis, inflammation, and immune escape.	[48,49,51,52,54,56,57,58,59]
Ovarian cancer is characterized by a different composition of viral signatures	*Anelloviridae*, *Astroviridae*, *Birnaviridae*, *Bornaviridae*, *Hepadnaviridae*, *Iridoviridae*, *Paramyxoviridae*, and *Rhabdoviridae* found at increased levels in ovarian cancer samples.	[55]
DNA of *Chlamydia* was found in 17% of ovarian cancer samples	*Chlamydia* infection provokes oxidative DNA damage, down-regulates the p53 function, and induces DNA double-strand breaks.	[67,69,70]
*Lactobacillus*-poor and highly diversified microbiota mimicking the post-menopausal environment were described in ovarian cancer	In healthy premenopausal women, a *Lactobacillus*-dominated microbiota was observed, while after menopause, highly diverse bacterial species dominated by *Propionibacterium*, *Corynebacterium*, and *Staphylococcus* were found.	[73,74]
*Proteobacteria* and *Firmicutes* were present in 52% and 22% of ovarian cancer samples, respectively. The *Proteobacteria*/*Firmicutes* ratio was significantly increased in ovarian cancer samples	Ovarian cancer samples had a much more diversified microbiota than controls.	[55,76]
*Firmicutes*, *Proteobacteria*, *Actinobacteria*, and *Bacteroidetes* phyla were identified in the upper genital tract of healthy women, while *Ralstonia*, *Mycobacterium*, *and Variovorax* were present in ovarian cancer	In the upper genital tract, the abundances of several bacterial taxa were reduced in cancer compared to control samples, indicating the occurrence of bacterial dysbiosis in ovarian cancer.	[79]
*Bacteroidales* in the gut microbiota might prevent the recolonization of cancer-associated *Prevotella* and further inhibit HGSOC development	There are connections between the gut microbiome and intestinal estrobolome (estrogen circulation and metabolism).	[29,82,83,84]
Genetic material from *Dipylidium*, *Trichuris*, *Leishmania*, *Babesia*, *Trichinella*, *Ascaris*, and *Trichomonas* was found in more than 95% of ovarian cancer samples	Parasitic signatures in ovarian cancer samples were significantly more diversified compared to controls.	[55]

**Table 2 biomedicines-12-00229-t002:** Accuracy of biomarkers and information about the studied groups.

Biomarker	Accuracy	Cases (N)	Reference
	**Mixed cases (FIGO I-II + III-IV) vs. controls**		
CA125	SP = 67% SE = 91% SP = 84% SE = 71% SP = 92% SE = 80% <50 years SP = 90% SE = 83% >50 years SP = 94% SE = 80% SP = 73% SE = 84% SP = 46% SE = 88% AUC 0.87 AUC 0.85—premenopausal AUC 0.92—postmenopausal	Patients N = 57 Controls N = 271 Patients N = 70 Controls N = 762 Patients N = 82 (<50 years-18, >50 years-64) Controls N = 1147 (<50 years-622, >50 years-525) Patients N = 2315 Controls N = 5875 Meta-analysis Patients N = 56 Controls N = 225 Patients N = 1862 Controls N = 4077 Meta-analysis	[185] [97] [186] [172] [173] [187]
HE4	SP = 97% SE = 57% SP = 87% SE = 71% SP = 97% SE = 71% SP = 76% SE = 90% <50 years SP = 88% SE = 100% >50 years SP = 60% SE = 87% SP = 90% SE = 73% SP = 98% SE = 53% AUC 0.89 AUC 0.85—premenopausal AUC 0.88—postmenopausal	Patients N = 57 Controls N = 271 Patients N = 2117 Controls N = 3837 Meta-analysis Patients N = 70 Controls N = 762 Patients N = 82 (<50 years-18, >50 years-64) Controls N = 1147 (<50 years-622, >50 years-525) Patients N = 2233 Controls N = 5682 Meta-analysis Patients N = 56 Controls N = 225 Patients N = 1862 Controls N = 4077 Meta-analysis	[185] [171] [97] [186] [172] [173] [187]
ROMA assay	SP = 91% SE = 90% SP = 97% SE = 71% SP = 81% SE = 88% <50 years SP = 75% SE = 100% >50 years SP = 87% SE = 84% SP = 80% premenopausal SP = 83% postmenopausal SE = 80% premenopausal SE = 88% postmenopausal SP = 88% premenopausal SP = 70% postmenopausal SE = 67% premenopausal SE = 89% postmenopausal SP = 80% SE = 97% AUC 0.86 AUC 0.84—premenopausal AUC 0.84—postmenopausal AUC 0.91 AUC 0.86—premenopausal AUC 0.93—postmenopausal	Patients N = 2117 Controls N = 3837 Meta-analysis Patients N = 70 Controls N = 762 Patients N = 82 (<50 years-18, >50 years-64) Controls N = 1147 (<50 years-622, >50 years-525) Patients N = 2315 Controls N = 5875 Meta-analysis Patients N = 56 Controls N = 225 Patients N = 100 Controls N = 132 Patients N = 74 Controls N = 186 Patients N = 1862 Controls N = 4077 Meta-analysis	[171] [97] [186] [172] [173] [188] [189] [187]
RMI assay	SP = 90% SE = 64% AUC 0.88 AUC 0.86—premenopausal AUC 0.88—postmenopausal	Patients N = 56 Controls N = 225 Patients N = 74 Controls N = 186	[173] [189]
CTCs DAPI+/CD45-/HE4+/EpCAM+/cytokeratin+/vimentin+	SP = 87% SE = 73%	Patients N = 61	[190]
cfDNA	AUC 0.89 At a specificity of 99.6%, required for ovarian cancer screening, the sensitivity of cfDNA testing was 2–5-fold higher compared with CA-125 and RMI testing	Patients N = 68 Controls N = 44	[191]
Expression of exosomal miR-93, miR-145 and miR-200c	miR-145 SP = 75% SE = 92% miR-200c SP = 90% SE = 73% CA125 + miR-145 SP = 60% SE = 98% CA125 + miR-200c SP = 70% SE = 94% miR-145 + miR-200c SP = 65% SE = 94% CA125 + miR-145 + miR-200c SP = 55% SE = 100%	Patients N = 68	[192]
Expression of exosomal miR-1290	SP = 87% SE = 69%	Patients N = 67 Controls N = 15	[193]
Expression of exosomal miR-99a-5p	SP = 75% SE = 85%	Patients N = 88 Controls N = 20	[194]
MCED test (Galleri ^TM^)	SP = 99% SE = 83%	Patients N = 8542 Controls N = 6712 Validation of the group of 3237 cancer and 2069 non-cancer patients	[189]
	**Advanced cases (FIGO III-IV) vs. controls**		
CA125	SP = 90% SE = 86% FIGO III SP = 90% SE = 94% FIGO IV	Serum: Patients N = 280 Controls N = 200 Plasma Patients N = 127 Controls N = 134 Monte–Carlo cross-validation method	[162]
HE4	SP = 86% SE = 89%	Patients N = 4549 Controls N = 8082 Meta-analysis	[101]
BARD1 autoantibody + CA125	SP = 92% SE = 94%	Serum: Patients N = 280 Controls N = 200 Plasma Patients N = 127 Controls N = 134 Monte–Carlo cross-validation method	[162]
PDLIM1 autoantibody	Distinguished 39.5% of patients with late-stage disease	Tissues: Patients N = 294 Controls N = 8 Serum: Patients N = 182 Controls N = 181 Training and validation set	[163]
CTCs	SE = 77% (+) in 73% of patients (+) in 82% of patients	Patients N = 44 Controls N = 43 Patients N = 66 Controls N = 5 Patients N = 160 Controls N = 90	[195] [196] [196]
ctDNA Methylation pattern of *RASSF1A*, *CALCA*, *EP300*	SP = 87% SE = 90%	Patients N = 30 Controls N = 30	[197]
ctDNA Methylation pattern of *RASSF1A*, *PGR-PROX*	SP = 73% SE = 80%	Patients N = 30 Controls N = 30	[197]
ctDNA Methylation pattern of 56 genes (group contained 87% of advanced HGSOC samples)	SP = 61% SE = 85%	Tissue: Patients N = 30 Controls N = 30 Plasma: Patients N = 33 Controls N = 33	[198]
	**Early cases (FIGO I-II) vs. controls**		
CA125	SP = 70–80% SE = 50–60% SP = 90% SE = 81%	Patients N = 456 Serum: Patients N = 280 Controls N = 200 Plasma Patients N = 127 Controls N = 134 Monte–Carlo cross-validation method	[92] [162]
HE4	SP = 87% SE = 64%	Patients N = 3570 Controls N = 245 Meta-analysis	[97]
OVCA1	Identified 63–78% of FIGO I-II patients who had normal CA125 levels	Patients N = 104 Controls N = 2201	[115]
Panel of ANXA2 + CA125	SP = 64% SE = 100%	Patients N = 160 Controls N = 143	[119]
ROMA assay	Indicated an 83% accuracy in diagnosing an early-stage disease	Patients N = 92 Controls N = 322	[185]
BARD1 autoantibody + CA125	SP = 92% SE = 95%	Serum: Patients N = 280 Controls N = 200 Plasma Patients N = 127 Controls N = 134 Monte–Carlo cross-validation method	[162]
PDLIM1 autoantibody	Distinguished 40.6% of patients with early-stage disease Identified 15% of patients who were negative with CA125	Tissues: Patients N = 294 Controls N = 8 Serum: Patients N = 182 Controls N = 181 Training and validation set	[163]
A panel of 6 antibodies + CA125	SP = 75% SE = 86%	Patients N = 300 Controls N = 200 Validation of detection models	[164]
CTCs	SE = 100% (+) in 10% of patients (+) in 74% of patients	Patients N = 44 Controls N = 43 Patients N = 66 Controls N = 5 Patients N = 160 Controls N = 90	[195] [199] [196]
Multiplex assay of aberrantly methylated 7 genes	SP = 90% SE = 85%	Patients N = 140 Controls N = 62 Validation of the assay on screening population	[200]
Combination of TRIM21, HARS, NYESO-1, PAX8, and TP53 markers	SP = 94% SE = 50%	Patients N = 114 Controls N = 50 Validation on independent serum set: Patients N = 100 Controls N = 50	[159]
A panel of 9 tumor-associated antigens (TAAs)	Identified 79% of patients who had normal CA125 levels	Patients N = 132 Controls N = 147	[160]
Panel of TTR, Hb, TF, and ApoAI markers	ROC 0.96	Patients N = 43 Controls N = 31	[117]
	**Prediagnostic cases vs. controls**		
CA125	In type II ovarian cancer CA125 was significantly up-regulated < 1 year tDx In type II ovarian cancer <1 year tDx SP = 73% SE = 90% In type II ovarian cancer 1–2 years tDx SP = 8% SE = 90% CA125 cut-off value 56.6 U/mL SP = 95% FIGO I: <6 months tDx SE = 43% <12 months tDx SE = 38% 1–2 years StDx SE = 14% FIGO III/IV: <6 months tDx SE = 93% <12 months tDx SE = 68% 1–2 years StDx SE = 26%	N = 30 type II ovarian cancer samples (FIGO I-7, FIGO II-8, FIGO III-15) Study group N = 30 patients/144 samples Controls N = 31 patients/158 samples EPIC Study group N = 810 ovarian cancer patients (FIGO III-IV-469) Controls N = 1939	[120] [121] [180]
HE4	HE4 cut-off value 39.4 pM SP = 95% FIGO I: <6 months tDx SE = 14% <12 months tDx SE = 19% 1–2 years StDx SE = 17% FIGO III/IV: <6 months tDx SE = 88% <12 months tDx SE = 72% 1–2 years StDx SE = 22%	EPIC Study group N = 810 ovarian cancer patients (FIGO III-IV-469) Controls N = 1939	[180]
CA72.4	CA72.4 cut-off value 2.5 U/mL SP = 95% FIGO I: <6 months tDx SE = 14% <12 months tDx SE = 25% 1–2 years StDx SE = 14% FIGO III/IV: <6 months tDx SE = 69% <12 months tDx SE = 56% 1–2 years StDx SE = 18%	EPIC Study group N = 810 ovarian cancer patients (FIGO III-IV-469) Controls N = 1939	[180]
CA15.3	CA15.3 cut-off value 1.4 mIU/mL SP = 95% FIGO I: <6 months tDx SE = 29% <12 months tDx SE = 19% 1–2 years StDx SE = 17% FIGO III/IV: <6 months tDx SE = 27% <12 months tDx SE = 16% 1–2 years StDx SE = 10%	EPIC Study group N = 810 ovarian cancer patients (FIGO III-IV-469) Controls N = 1939	[180]
Panel of LCAT, IGFBP2, SHBG, and CA125	For type II ovarian cancer, SHBG was significantly down-regulated at 2–3 years tDx For type II ovarian cancer, the SHBG:IGFBP2:CA125 model significantly outperformed CA125 at 1–2 and >4 years tDx For type II ovarian cancer, the IGFBP2:LCAT:CA125 model significantly outperformed CA125 at 1–2 and >4 years tDx For type II ovarian cancer, the SHBG:IGFBP2:LCAT:CA125 model significantly outperformed CA125 at 1–2 and >4 years tDx For type II ovarian cancer, the IGFBP2:LCAT:CA125 model lead time was 272 days tDx compared to CA125 alone (165 days tDx)	N = 30 type II ovarian cancer samples (FIGO I-7, FIGO II-8, FIGO III-15)	[120]
Panel of CHI3L1, AGR2, CA125 and HE4	CA125: AGR2: CHI3L1 and CA125: CHI3L1: HE4 models In type II ovarian cancer <1 year tDx SP = 100% SE = 90% In type II ovarian cancer 1–2 years tDx SP = 20–24% SE = 90%	Study group N = 30 patients/144 samples Controls N = 31 patients/158 samples	[121]
CA125, HE4, mesothelin	Serum concentrations began to increase approximately 3 years before diagnosis, but detectable elevations were noticed 1 year before diagnosis For CA125, the AUC was 0.57 >4 years tDx 0.68 2–4 years tDx 0.74 <2 years tDx	CARET cohort N = 18,314 (6289 postmenopausal women) 35 participants (+) for ovarian cancer Controls N = 70	[174]

**Table 3 biomedicines-12-00229-t003:** The usefulness of liquid biopsy for diagnosis of ovarian cancer.

	**Circulating Tumor Cells (CTCs)**	
**Phenotype of CTCs**	**Results of Testing**	**Reference**
EpCAM+	Positive in peripheral blood of 12% of ovarian cancer patients	[208]
CK7+ CK8+ CK18+ CK20+ TFS-2+ EGFR+	Positive in peripheral blood of 19% of ovarian cancer patients, especially advanced grade 3 tumors	[209]
EpCAM+ CK4+ CK5+ CK6+ CK8+ CK10+ CK13+ CK18+	Positive in peripheral blood of 10% of early-stage ovarian cancer patients and 73% of advanced tumors	[199]
EpCAM+ MUC-1+ HER-2+ CA125+	Positive in peripheral blood of 19% of ovarian cancer patients	[210]
EpCAM/CK+ combined with cell size and deformability	Positive in peripheral blood of 56% of ovarian cancer patients, including 100% of early-stage and 68% of late-stage ovarian cancer patients and 44% of benign ovarian tumor patients	[195]
Hoechst+ CK+ CD45-	Showed counts of more than 8 cells in 87% of ovarian cancer patients but in none of the healthy controls	[211]
EpCAM+ CK+ CD45- HE4+ Vimentin+	Positive in peripheral blood of 49% of ovarian cancer patients	[190]
EpCAM+ FRα+	The capture rate was 92% using the combination of EpCAM+FRα+, exceeding the rate when both markers were used alone by 20%	[216]
EpCAM+ MUC1+ WT1+	Significantly higher specificity than CA125 (92% vs. 82%), especially in early-stage ovarian cancer (74% vs. 58%)	[218]
	**Circulating tumor DNA (ctDNA)**	
**Studied Genes**	**Results of Testing**	**Reference**
*OPCML*	The methylation of *OPCML* showed a significant difference between early-stage ovarian cancer patients and healthy controls	[227]
*RASSF2A*	Inactivation of *RASSF2A* through aberrant promoter methylation was found in 36% of ovarian cancer patients	[228]
*RASSF1A*, *RUNX3*, *TFPI2*, *OPCML*, *SFRP5*, *CDH1*, *APC*	The multiplex assay of aberrantly methylated genes achieved a sensitivity of 85% and a specificity of 90% in early-stage ovarian cancer	[200]
*RASSF1A*, *CALCA*, *EP300*	Methylation of these genes enabled differentiation between ovarian cancer patients and healthy controls with a sensitivity of 90% and a specificity of 87%	[197]
*RASSF1*, *CDH1*, *PAX1*, *PTEN*	Aberrant hypermethylation was found for *CDH1* and *PAX1* genes in malignant ovarian lesions	[231]
*RASSF1A*, *PGR-PROX*	Methylation pattern discriminated between ovarian cancer and benign tumors with a sensitivity of 80% and a specificity of 73%	[197]
*SFRP1*, *SFRP2*, *SFRP4*, *SFRP5*, *SOX1*, *PAX1*, *LMX1A*	The methylation pattern showed a sensitivity of 75% and a specificity of 73% as a screening marker	[230]
Cancer SEEK test	Diagnosis of 8 cancers, including ovarian cancer with a sensitivity of 69–98% and a specificity of 99%	[233]
Pap SEEK test	Mutations and aneuploidy of 18 genes enabled the identification of 33% of patients; when combined with an intrauterine brush it identified 45% and when combined with plasma ctDNA it identified 63% of ovarian cancer patients, respectively	[238]
*TP53*	In ovarian cancer patients, the allelic frequencies of 2–65% were found in blood samples with sensitivity and specificity of more than 97%	[191]
	The presence of mutation was confirmed in 60% of uterine lavage samples of patients with ovarian cancer	[239]
	Similar results were obtained when *TP53* mutations were found in DNA extracted from vaginal tampons	[240]
Chromosomal instability	Improved malignancy detection (AUC 0.94) compared to CA125 (AUC 0.78) and RMI index (AUC 0.81)	[191]
NIPT sequencing	Detected 41% of all HGSOC, including 38% of early-stage cases	[236]
CNI-score	A sensitivity of 91% and a specificity of 95% to detect ovarian cancer in blood samples	[237]
	**Exosomes**	
**Exosome Load**	**Results of Testing**	**Reference**
*FGA*, *GSN*, *FGG*, *LBP*	Over-expression of *FGA* and *GSN* and under-expression of *FGG* and *LBP* in ovarian cancer patients	[193]
miRNAs	miR-21, miR-100, miR-200b, and miR-320 were significantly increased, whereas miR-16, miR-93, miR-126, and miR-223 were decreased in plasma exosomes of ovarian cancer patients as compared to healthy women	[194]
	miR-93, miR-145, and miR-200c showed higher expression in serum exosomes of the HGSOC group, compared to borderline and benign tumors	[244]
	miR-1290 was significantly over-expressed in serum exosomes of malignant ovarian tumors compared to benign tumors	[246]
	Exosomal level of miR-99a-5p was up-regulated in the sera of ovarian cancer patients	[247]
Proteins	L1CAM, CD24, ADAM10, EMMPRIN, CLDN-4, TGF-β were up-regulated in sera of ovarian cancer patients	[248,249,250]
ExoProfile chip	FRα, CA125, EpCAM, and CD24 were able to individually detect some controls, early-stage, and advanced ovarian cancer, but the combined set outperformed the individual markers and differentiated all three groups significantly	[252]

EpCAM—epithelial cell adhesion molecule; CK—cytokeratin; TFS-2—transcription factor-2; EGFR—epithelial growth factor receptor; MUC-1—mucin-1; HER-2—human epidermal growth factor receptor-2; CA125—cancer antigen 125; WT-1—Wilms tumor protein-1; *OPCML*—opioid binding protein/cell adhesion molecule like; *RASSF2A*—Ras-association domain family 2A; *RUNX3*—Runt-related transcription factor 3; *TFPI2*—tissue factor pathway inhibitor 2; *SFRP5*—secreted frizzled-related protein 5; *CDH1*—cadherin 1; *APC*—APC regulator of WNT signaling pathway; *CALCA*—calcitonin related polypeptide alpha; *EP300*—adenovirus E1A-associated cellular p300 transcriptional co-activator; *PAX1*—paired box gene 1; *PTEN*—PTEN phosphatase and tensin homolog; *PGR-PROX*—progesterone receptor-Prospero homeobox protein 1; TP53—tumor protein p53; RMI—risk of malignancy index; NIPT—non-invasive prenatal testing; HGSOC—high-grade serous ovarian cancer; CNI-score—copy number instability score; miRNA—micro ribonucleic acid; L1CAM—L1 cell adhesion molecule; CD24—heat-stable antigen CD24; ADAM10—a disintegrin and metalloproteinase domain-containing protein 10; EMMPRIN—extracellular matrix metalloproteinase inducer; CLDN-4—claudin-4; TGF-β—transforming growth factor beta; FRα—folate receptor-alpha.

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
