# Peer review of "High-Grade Serous Ovarian Cancer—A Risk Factor Puzzle and Screening Fugitive"

_biomedicines, 2024, doi:10.3390/biomedicines12010229_

Round 1

Reviewer 1 Report

Comments and Suggestions for Authors

This paper is a useful review of issues involving detection of high grade serous ovarian cancer. It extensively covers this topic with sections on many different marker and marker combinations. It concludes with a future direction section. The paper does need much work on English grammar and punctuation. I recommend a through editing by someone proficient in English. This would make the paper much more valuable for the reader.

Comments on the Quality of English Language

The paper needs a thorough editing to correct many issues with English grammar and punctuation. 

Author Response

# PT Reviewer No 1

This paper is a useful review of issues involving detection of high grade serous ovarian cancer. It extensively covers this topic with sections on many different marker and marker combinations. It concludes with a future direction section. The paper does need much work on English grammar and punctuation. I recommend a through editing by someone proficient in English. This would make the paper much more valuable for the reader.

Thank you very much for your positive opinion, time to complete the review and valuable remarks on language correction. We profoundly appreciate it. 

 The English grammar has been checked and improved by the Grammarly Editor

Thank you, kind regards

Authors

Reviewer 2 Report

Comments and Suggestions for Authors

A very comprehensive and well-referenced review of risk factors and predictive markers for HGSOC.

The structure is on the whole fine, although the section on risk factors would have been helped by the inclusion of (a) summary table(s) comparing and contextualising the different RFs together. The conclusions could also be more definitive, with a more detailed explanation and justification given for the 'DEPHENCE' system.

Comments on the Quality of English Language

There are several grammatical errors throughout which need to be addressed.

Author Response

# PT Reviewer No 2

A very comprehensive and well-referenced review of risk factors and predictive markers for HGSOC.

We very much appreciate your positive opinion, time to complete the review and valuable remarks which helped us to improve the manuscript.

The structure is on the whole fine, although the section on risk factors would have been helped by the inclusion of (a) summary table(s) comparing and contextualising the different RFs together. The conclusions could also be more definitive, with a more detailed explanation and justification given for the 'DEPHENCE' system.

The table summarizing the risk factors for HGSOC with short comments has been included

The more detailed explanation and justification for the „Dephence” system has been supplemented

 Comments on the Quality of English Language - There are several grammatical errors throughout which need to be addressed.

 The English grammar has been checked and improved by the Grammarly Editor

Thank you, kind regards

Authors

Reviewer 3 Report

Comments and Suggestions for Authors

This is a comprehensive and timely review. To be useful for readers it needs substantial improvements in regard to organization and critical evaluation of data presented. Specifically:

1.       Typos and grammatical errors

2.       No description of histotypes and precursors: STIC is only for HGSC and only for subtype originating from FT.

3.       Requirements for early detection of OC in terms of specificity, sensitivity, and PPV should be presented.

4.       Interventional relevance of gene polymorphisms and microbiome is not suggested.

5.       List of prospective studies does not include Shizuoka trial.

6.       ROMA and like tests should be grouped with IOTA along with the statement that these tests are for differential diagnosis of ovarian cancer that are different from early detection.

7.       The authors also mention using biomarkers (iCTC) for disease monitoring, which is different from early detection and differential diagnosis. All these potential usages of biomarkers in ovarian cancer should be mentioned and spelled out in the introduction, and presented separately in the body of review.

8.       Validated biomarkers should be reported separately from biomarkers not validated in independent blinded sets. Additionally, number of cases and controls used in each study should be considered to report only significant findings devoid of potential overfitting.

9.       Candidate biomarkers developed in late case vs control studies should be reported separately from those developed in early cases vs control and in pre-diagnostic cases vs control studies.

10.   Limited diagnostic power of CA125 in samples collected >1 yr before clinical diagnosis should be noted.

11.   In reference 161 expression of PAX8 in fallopian tubes’ secretory cells should be noted.

12.   New York esophageal squamous cell carcinoma 1 was renamed to cancer testis antigen CTAG1 (since later CTAG2 is mentioned).

13.   Mortality reduction in UKCTOCS trial did not reach statistical significance (Ref 108).

14.   Line 666 – sensitivity is also presented and should be mentioned.

15.   Grail methylation test, its performance in prospective study, and Galleri company should be mentioned.

16.   Since CTCs and cfDNA biology is briefly discussed, exosome biology should be also briefly discussed.

17.   IOTA algorithm is included in population screening, however, it was initially developed for differential diagnosis.

Comments on the Quality of English Language

English language should be edited

Author Response

# PT Reviewer No 3

This is a comprehensive and timely review. To be useful for readers it needs substantial improvements in regard to organization and critical evaluation of data presented. 

We very much appreciate your positive opinion, time to complete the review and valuable and detailed remarks which helped us to improve the manuscript.

1.       Typos and grammatical errors

 The English grammar has been checked and improved by Grammarly Editor

2.       No description of histotypes and precursors: STIC is only for HGSC and only for subtype originating from FT.

The alternative pathway for genesis of HGSOC tumors has been shortly described in the „Introduction” section.

3.       Requirements for early detection of OC in terms of specificity, sensitivity, and PPV should be presented.

The requirements for early detection of OC has been cited and shortly discussed 

4.       Interventional relevance of gene polymorphisms and microbiome is not suggested.

The intervention connected with the microbiome has been shortly suggested. The interventional relevance of SNPs has been added.

5.       List of prospective studies does not include Shizuoka trial.

Shizuoka trial has been cited

6.       ROMA and like tests should be grouped with IOTA along with the statement that these tests are for differential diagnosis of ovarian cancer that are different from early detection.

The appropriate parts of the text have been reorganized

7.       The authors also mention using biomarkers (iCTC) for disease monitoring, which is different from early detection and differential diagnosis. All these potential usages of biomarkers in ovarian cancer should be mentioned and spelled out in the introduction, and presented separately in the body of review.

The potential use of biomarkers in disease monitoring has been mentioned in the „Introduction”. Moreover, the separate chapter devoted to biomarkers in cancer monitoring has been added. 

8.       Validated biomarkers should be reported separately from biomarkers not validated in independent blinded sets. Additionally, number of cases and controls used in each study should be considered to report only significant findings devoid of potential overfitting.

The information about validation has been added in the new table 2., similarly like number of studied patients

9.       Candidate biomarkers developed in late case vs control studies should be reported separately from those developed in early cases vs control and in pre-diagnostic cases vs control studies.

The new table 2 contains separate data concerning the mixed (early and advanced together), advanced, early and pre diagnostic cases vs. controls

10.   Limited diagnostic power of CA125 in samples collected >1 yr before clinical diagnosis should be noted.

It has been noted both in the text and in the table 2.

11.   In reference 161 expression of PAX8 in fallopian tubes’ secretory cells should be noted.

It has been noted

12.   New York esophageal squamous cell carcinoma 1 was renamed to cancer testis antigen CTAG1 (since later CTAG2 is mentioned).

The name has been up-dated

13.   Mortality reduction in UKCTOCS trial did not reach statistical significance (Ref 108).

The original text was slightly modified: „The UK Collaborative Trial of Ovarian Cancer Screening (UKCTOCS) revealed that longitudinal CA125 testing combined with annual TVUS screening improved the recognition capability for early-stage ovarian cancer by 47%, but it was not accompanied with significant decrease in mortality” 

14.   Line 666 – sensitivity is also presented and should be mentioned.

It was supplemented

15.   Grail methylation test, its performance in prospective study, and Galleri company should be mentioned.

The paper : E. A. Klein et al. „Clinical validation of a targeted methylation-based multi-cancer early detection test using an independent validation set” has been cited

16.   Since CTCs and cfDNA biology is briefly discussed, exosome biology should be also briefly discussed.

Exosome biology has been briefly described on the beginning of the „Exosomes” sub-chapter

17.   IOTA algorithm is included in population screening, however, it was initially developed for differential diagnosis.

It was placed in new sub-chapter entitled „2.2 Assessment of adnexal mass - prediction of ovarian cancer in symptomatic patients”

Thank you, kind regards

Authors

Round 2

Reviewer 1 Report

Comments and Suggestions for Authors

Corrections and additions are appropriate

Author Response

Dear P.T. Reviewer,

Thank you very much for your positive opinion. We appreciate very much your time and valuable cooperation which has helped to improve our manuscript.

Best regards

Authors

Reviewer 2 Report

Comments and Suggestions for Authors

The authors have addressed my concerns with the content appropriately. Think it still needs some significant checking of the paper for grammatical errors.

Comments on the Quality of English Language

The language still needs revision in places.

Author Response

Dear P.T. Reviewer,

Thank you for your time and valuable remarks that have helped us considerably to improve our manuscript. According to your remarks the English language has been extensively corrected both in style and grammar. The corrections have not been marked due to the number (more than 350). We appreciate very much your valuable contribution to the improvement of the text.

Best regards

Authors